# Multi-session smart night charging of EVs for accelerated decarbonization of electric mobility

**Felix Wieberneit**[1]*, **Emanuele Crisostomi**[2], **Anthony Quinn**[1,3],
**Homayoun Hamedmoghadam**[1], **Robert Shorten**[1]

**1** Dyson School of Design Engineering, Imperial College London, London, United Kingdom, **2** Department of Energy, Systems, Territory and Constructions Engineering, University of Pisa, Pisa, Italy, **3** Department of Electronic and Electrical Engineering, Trinity College Dublin, Dublin, Ireland

* fw1520@ic.ac.uk

## Abstract

The majority of electric vehicles (EVs) are charged domestically overnight, when the precise timing of power allocation is not important to the user, thus providing a source of flexibility that can be leveraged by charging control algorithms. In this paper, we show that by keeping the EV connected to a standard domestic smart plug every night, it is possible to achieve an approximate 37% annual reduction in carbon intensity compared to uncontrolled charging (based on 2022 UK National Grid data) without compromising the EV owners' charging demand. For this purpose, we use Model Predictive Control techniques to schedule power delivery with the objective of minimizing grid-average $CO_2$ emissions over forecast windows of multiple sessions (up to seven days ahead). We find that flexibility on the part of EV owners—both their willingness to keep the EV connected at night, and to indicate a precautionary (i.e., upper bound) estimate of their required state-of-charge in the morning—allows for significant emissions reduction and accelerated decarbonization of the mobility sector.

## 1 Introduction

Electric Vehicles (EVs) are meeting the expectation of significantly reducing greenhouse gas emissions, as their Global Warming Potential (GWP), expressed in $gCO_2e/km$ (grams of carbon dioxide equivalent per vehicle-kilometre), is typically much lower than that of gasoline and diesel Internal Combustion Engine Vehicles (ICEVs) [1]. For instance, the net lifecycle impact of new EVs in 2020 is lower than that of new gasoline and diesel cars in all European countries, except for Estonia. Based on EV lifecycle assessments, in most European countries, charging events account for the most significant share of the total lifecycle GWP.

While production, maintenance, and non-use phase emissions are largely embodied (i.e., emissions are emitted, even before the EV is used) and relatively country-invariant, the use-phase GWP from charging is highly contingent on the specific power generation mix, which varies in time and space. Consequently, the same EV

**Data availability statement:** The data underlying the findings in this manuscript have been published in the public Zenodo repository, accessible under https://doi.org/10.5281/zenodo.17287590.

**Funding:** FW and RS acknowledge support from the IOTA Foundation for this work.

**Competing interests:** The authors have declared that no competing interests exist.

yields markedly different lifecycle GWP depending on the carbon intensity of the power generation $C_{gen}$ used to charge it: GWP from charging can be high in countries with predominantly fossil-based grids (e.g., Estonia) and low in countries with predominantly decarbonized grids (e.g., France or Sweden) [1].

While $C_{gen}$ is relatively low in many European countries, it remains substantially higher in regions with fossil-dominated generation mixes, and the world-average $C_{gen}$ is almost double that of Europe [1]. Power generation from renewable sources is increasing worldwide, and as a consequence, $C_{gen}$ is expected to decrease. In the meantime, however, smart charging strategies may accelerate the decarbonization of EVs, by aligning demand with periods of lower $C_{gen}$, effectively using EVs as mobile storage systems, exploiting their flexibility to integrate fluctuating renewables by absorbing surplus at night or during high-wind/low-load conditions.

Unidirectional charging control or "smart charging" is widely recognized as the most convenient way to accommodate the electrification of the mobility sector while mitigating the need for electricity network expansion [2]. Roughly speaking, smart charging refers to the possibility to control the power that is used to charge a plugged EV, as an alternative to uncontrolled charging, where the EV is charged at maximum power from the moment it is connected, until it is fully charged. Smart charging has been investigated extensively in the literature for a plethora of applications, and most notably for load balancing [3,4]; peak shaving [5,6]; energy cost minimization [7,8]; grid impact minimization [9]; or for minimizing waiting times [10,11]. A comprehensive review of smart charging strategies can be found in Ref. [12], and, with a focus on distributed solutions in Ref. [13]. Many references also explicitly focus on fleets of EVs, as more flexibility can be gained through collective control of a (large) number of EVs [14].

Nevertheless, studies explicitly targeting carbon emission minimization remain comparatively limited. Early work demonstrates that aligning charging with low-carbon periods or renewable availability can yield substantial reductions: Dixon et al. [15] showed fleet-scale scheduling in Scotland reduced emissions by up to 20–30% while absorbing curtailed wind, and others [16,17] emphasized the importance of marginal emission factors for accurate $CO_2$ savings estimates. Recent contributions have combined carbon objectives with time-of-use pricing signals [18], or examined systemic interactions with renewable penetration thresholds [19]. However, most strategies assume perfect foresight [18], operate at the fleet/aggregator scale [15], or optimize over short horizons (i.e., a single charging session, or a single day) [15,16]. Studies on grid carbon intensity forecasting highlight that machine learning based approaches can be used to forecast grid carbon intensity with high accuracy over short time horizons (up to six hours ahead [20], while recently, the feasibility of longer horizon forecasts (up to 4 days ahead) has been demonstrated, with mean absolute percentage errors of 5–14% (depending on region and growing with horizon length) [21]. Other recent work has emphasized probabilistic forecasts: Peng et al. [22] demonstrate a forecasting method that yields uncertainty intervals, while Ref. [23] provide a review on the statistical metrics used to quantify uncertainty in carbon intensity forecasts. These contributions show that current machine learning

approaches enable forecasting of grid carbon intensity multiple days ahead. Forecast uncertainty tends to increase with forecast horizon and differs between methods, with some methods explicitly yielding uncertainty intervals. Regional variations in forecast uncertainty are also evident [21]. Such considerations are to be taken into account in the design of charging control systems that rely on predicted carbon intensity signals.

Contrary to the prior literature, our study focuses on the under-explored domestic overnight charging case, requiring only a simple smart plug. We introduce a multi-session Model Predictive Control (MPC) framework that extends the optimization horizon across several days (up to one week), explicitly accounting for uncertainty in long-horizon carbon intensity forecasts and user behaviour. This approach allows us to leverage temporal variability in grid carbon intensity beyond single-night horizons for emission reduction and we demonstrate that up to 37% emission reductions (UK National Grid, 2022) are achievable with minimal hardware requirements. In this paper, the UK National Grid is selected as a case study because it provides rich, publicly available half-hourly carbon intensity data. The proposed MPC method can be simply applied to grids with differing $C_{gen}$ sequences, as long as $C_{gen}$ is not constant in time, however, naturally, the achievable emission reductions will differ from those reported here for the UK National Grid as emission reductions are contingent on the specific $C_{gen}$ sequence.

In summary, we extend the existing literature along the following directions:

1. Firstly, in contrast to prior studies, which focus on fleet-scale, aggregator-led approaches, we reconsider the unidirectional smart charging problem for a single EV in a domestic overnight scenario, involving only a simple smart plug. We demonstrate that significant emission reductions are possible with minimal hardware and communication requirements;

2. Secondly, while existing works optimize power allocation during a single charging session, we solve the optimization problem for a sliding forecast window spanning multiple charging sessions (e.g., four days or one week), by applying predictive control techniques [24]. In particular, this enables us to leverage the additional flexibility to delay charging of surplus energy to subsequent nights. Our simulations do not rely on the assumption of perfect carbon intensity forecasts. Instead, we take into account linearly increasing uncertainty of forecasts with forecast horizon. Our results highlight that multi-session optimization can significantly enhance emission reductions;

3. Thirdly, our simulations explicitly quantify how EV driver behaviour, such as plug-in times and duration and daily energy consumption, moderate the achievable emission reductions, with implications for the design of behavioural incentives or market-based mechanisms for the trading of emission reductions;

4. Finally, using regional carbon intensity data from 14 different UK regions, we assess the emission reduction potential of the proposed method. We highlight that absolute emission reductions are particularly relevant in regions with high carbon intensity.

This paper is organized as follows: in the next section, we formulate the carbon intensity minimization problem, and describe the MPC strategy for overnight charging control. A list of symbols can be found in Table 1. In Section 3, we explain how to model the uncertain variables of the optimization problem, most notably future carbon intensity of energy, and driving patterns of EV users. In Section 4, through numerical simulations we evaluate the effectiveness of the proposed charging strategy, and furthermore investigate the effect of different forecasting horizons and varying charging flexibility on the potential for emission reduction. The results are concluded by highlighting regional differences in terms of emission reduction potential. Finally, in Section 5 we discuss the main findings together with possible future directions to expand and address the limitations of this work.

## 2  The MPC-based charging strategy

Fig 1 shows the carbon intensity of electricity generation, $C_{gen}$ (gCO$_2$e/kWh), in the UK during the first week of January 2022. This sequence is made available half-hourly by the National Energy System Operator (NESO) [25]. Preliminary inspection reveals the characteristic daily pseudo-periodicity of this time series. In particular, $C_{gen}$ significantly changes in

**Table 1. List of symbols.**

| Symbol | Meaning |
| --- | --- |
| $C_{gen}[l]$ | Actual grid carbon intensity at interval $l$ (gCO$_2$/kWh) |
| $\hat{C}_{gen}[l]$ | $l$-step-ahead forecast of $C_{gen}$ |
| $\hat{C}_{gen}^{(1)}[l]$ | Published 1-step-ahead forecast (NESO) |
| $\hat{\epsilon}[l]$ | Relative $l$-step forecast error |
| $N$ | Forecast window length (days) |
| $c$ | Intervals per day ($=24/\Delta_t$) |
| $P[s,k]$ | Grid-side charging power at day $s$, interval $k$ (kW) |
| $SOC[s,k]$ | State of charge at day $s$, interval $k$ (%) |
| $B$ | Battery capacity (kWh) |
| $\eta$ | Charging efficiency (grid-to-battery), $\eta \in (0,1]$ |
| $\hat{E}_s$ | Estimated next-day energy need (kWh) |
| $\underline{SOC}, \overline{SOC}$ | SOC bounds (%) |
| $SOC_s$ | Required morning SOC for day $s$ (%) |
| $\hat{k}_{s,b}, \hat{k}_{s,e}$ | Start/end indices of charging session during day $s$ |
| $\Delta_t$ | time step duration (here, $\Delta_t = 0.5$ h) |
| $\hat{\lambda}$ | Carbon intensity forecast error growth rate per interval |
| $C_{EV}$ | Average carbon emissions of energy charged by the EV (gCO$_2$/kWh) |

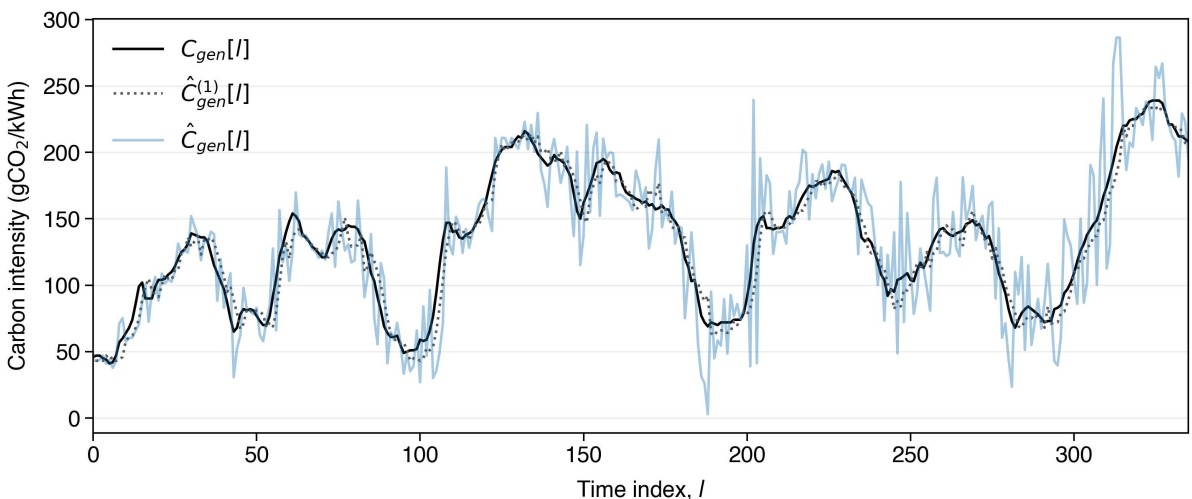

**Fig 1. $C_{gen}[l]$, $l \in \{1,\ldots,48\times7\}$ (black line):** *actual* half-hourly carbon intensity measurements for UK National Grid electricity generation during the first week of January 2022, published by the National Energy System Operator (NESO) [25]; $\hat{C}_{gen}^{(1)}[l]$ **(black dotted line):** NESO's published sequence of *one-step-ahead* forecasts; $\hat{C}_{gen}[l]$ **(blue line):** our *l-step-ahead* forecast sequence (12), estimated by transferring Carbon-Cast side-information [21].

time, reaching values that are approximately five times higher than the minima observed during the week. Carbon intensity depends on the electric load: when consumption is greater, then more fossil-based power generation is required to supplement generation from renewable sources, and so carbon intensity increases. Correspondingly, $C_{gen}$ tends to be

lower during the night, when the electric load is lower, and so generation from renewable sources (e.g., especially wind) can satisfy most of the electrical energy demand. In addition to the daily pseudo-periodicity, we observe significant variation between days. The significant temporal variability of carbon intensity suggests that controlled EV charging—in which charging is scheduled during periods of low $C_{gen}$—has the potential to significantly reduce the carbon intensity associated with EV usage [17,18].

## 2.1 The predicted net $CO_2$ objective function, $\hat{J}_0$

We adopt an MPC strategy [24] which schedules power delivery to the EVs during low-carbon overnight charging intervals. We want to optimize this schedule in the context of the predicted variation in carbon intensity, $\hat{C}_{gen}$, over multiple nights. Since a typical driver may not deplete the entire battery capacity during a day's driving, there is flexibility to delay charging to later (i.e., longer-horizon) nights, if $\hat{C}_{gen}$ is predicted to be high during the next (i.e., shorter-horizon) nights. Our predictive net $CO_2$ objective function should therefore meet the following criteria:

- minimize the net $CO_2$ emissions of charging the EV battery over a planning horizon—which we call the *forecast window*—of several nights' duration;

- ensure that the SOC of the battery remains within safe operational limits at all times (assumed to be between 20% and 80% in this paper);

- ensure that the battery is sufficiently charged every morning to cover that day's predicted driving needs.

The MPC strategy then iteratively solves the minimization problem for net $CO_2$ over a sliding forecast window.

**2.1.1 Discretization of the forecast window.** Let the forecast window comprise $N \geq 1$ (24-hour) days, each divided into $c \equiv \frac{24}{\Delta_t}$ contiguous intervals of duration $\Delta_t$ ($\equiv 0.5$ hours in our implementation), and indexed by $l \in \{1, \ldots, c \times N\}$. The minimization of net $CO_2$ depends, of course, on $\hat{C}_{gen}[l]$, the predicted carbon intensity sequence for electricity generation during the forecast window, which is assumed to be constant during each $\Delta_t$ interval. Since these predictions are only required during the active (presumably overnight) charging sessions, it is convenient also to adopt a double-indexing convention,

$$\hat{C}_{gen}[s, k] \equiv \hat{C}_{gen}[l]\Big|_{l \equiv c \times (s-1) + k},$$

$$(1)$$

where $s \in \{1, \ldots, N\}$ and $k \in \{1, \ldots, c\}$; i.e., $\hat{C}_{gen}[l]$ can also be indexed by the $k$ interval within the $s$th day of the forecast window.

**2.1.2 Constrained minimization of net $CO_2$.** We denote by $\hat{J}[l = 0] \equiv \hat{J}_0$ the predicted net $CO_2$ (g$CO_2$) associated with the multiple (typically overnight) charging sessions. Our objective is to minimize $\hat{J}_0$ by choosing an optimal sequence of charging powers, $\{P[s,k]\}$ (kW), during the $N$ charging sessions of the forecast window:

$$\min_{\{P[s,k]\}} \hat{J}_0 \equiv \sum_{s=1}^{N} \sum_{k=\hat{k}_{s,b}}^{\hat{k}_{s,e}} \hat{C}_{gen}[s, k] \times P[s, k] \times \Delta_t.$$

$$(2)$$

On the $s$th 24-hour day (but typically during the night), the charging session will begin in the $\hat{k}_{s,b}$th interval, and end in the $\hat{k}_{s,e}$th interval, $\hat{k}_{s,e} > \hat{k}_{s,b}$, $\forall s$. $P[s,k]$ (kW) denotes the sequential charging powers during each $\Delta_t$-interval of each overnight charging session. These are again assumed to be constant over the intervals of duration, $\Delta_t$, and their optimization for minimal net $CO_2$, $\hat{J}_0$, is the purpose of our scheme. The optimization (2) is subject to technology constraints associated with the domestic charging infrastructure and the EV battery itself. The SOC sequence of the EV battery (in %) satisfies the following update equations:

$$SOC[s, k + 1] = SOC[s, k] + \frac{\eta \times P[s, k] \times \Delta_t \times 100}{B},$$
$$k \in \{\hat{k}_{s,b}, \ldots, \hat{k}_{s,e} - 1\}, \tag{3}$$

$$SOC[s + 1, \hat{k}_{s+1,b}] = SOC[s, \hat{k}_{s,e}] - \frac{\hat{E}_s \times 100}{B},$$
$$s \in \{1, \ldots, N - 1\}. \tag{4}$$

Here, $B$ denotes the battery capacity (kWh). (3) models the increasing SOC sequence due to charging during the $s$th charging session, with $P[s, k]\Delta_t$ denoting the grid energy delivered in interval $(s,k)$, and $\eta P[s, k]\Delta_t$ denoting the energy stored in the battery. Eq. (4) expresses the reduction in SOC between successive days, due to $\hat{E}_s$, being the estimated energy consumed from the battery because of the use of the EV during the $s$th day. Finally, we can express the constraints imposed on the optimization problem (2), both directly in terms of $P[s,k]$, and indirectly in terms of $SOC[s,k]$:

$$0 \leq P[s, k] \leq \overline{P}, \tag{5}$$

$$\underline{SOC} \leq SOC[s, k] \leq \overline{SOC}, \tag{6}$$

$$SOC[s, \hat{k}_{s,e}] \geq \underline{SOC}_s. \tag{7}$$

In (5), $\overline{P}$ is the nominal maximum power rating (kW) of the domestic infrastructure. Meanwhile, $(\underline{SOC}, \overline{SOC}) \equiv (20, 80)\%$ (6) is the interval within which the SOC should be maintained. These bounds are chosen to reflect commonly recommended operating ranges intended to reduce degradation in practical operation, and this is in line with recent reports [2], which estimate 40% increases in battery lifetime by capping charging targets at 80%. Finally, $\underline{SOC}_s$ in (7) is the minimum SOC required at the end of the $s$th overnight charging session in order to meet the energy consumption, $\hat{E}_s$ (kWh) (4), of the day ahead. The net $CO_2$ objective, $\hat{J}_0$ (2), depends, not only on the predicted carbon intensity sequence, $\hat{C}_{gen}[l]$, during the charging sessions, i.e., $\hat{C}_{gen}[s, k]$ (1), but also on specification of the start and end times of each of these sessions, via $\hat{k}_{s,b}$ and $\hat{k}_{s,e}$, respectively. It also depends on the estimated energy consumption sequence, $\hat{E}_s$, via (4). Our technique for computing $\hat{C}_{gen}[l]$ from an external database will be explained in Section 3.1. Our computation of $\hat{k}_{s,b}$ and $\hat{k}_{s,e}$ will be explained in Section 3.2, and of $\hat{E}_s$ in Section 3.3.

**Remark (Model Accuracy):** In this work we have not considered the electrochemical dynamics of the charging process, and we have neglected aspects such as SOC-dependent charging/discharging efficiency, or voltage-recovery corrections, or battery ageing. Related to this last aspect, we remind however that a recent report [2] has found that smart charging strategies in general, as the one we are advocating in this manuscript, may further extend the battery lifetime by about 5–10% with respect to uncontrolled charging. While in this manuscript the main focus of our work is the comparison between uncontrolled and smart charging, more accurate models may be considered in a future work for more realistic modelling. We treat the charging efficiency parameter $\eta$ as constant for domestic AC charging over the SOC range considered (20–80%), and we assume the same $\eta$ across all compared strategies; under this standard approximation, $\eta$ primarily rescales SOC evolution and absolute energy flows, and does not materially affect the relative comparison between uncontrolled charging, single-session MPC, and multi-session MPC. The proposed method is hardware-agnostic: it requires only an interface to actuate charging power (continuously via a controllable EVSE, or approximately via time-averaged on/off switching, as can be implemented with smart plugs). As such, the model can be extended with more realistic, hardware specific modelling of charging efficiency.

**Remark (Computational Performance):** Each MPC step solves a convex LP with $N \times c$ decision variables (e.g., $N = 4$, $c = 48 \rightarrow 192$ variables) and $\mathcal{O}(Nc)$ linear constraints. On an Apple M2, 8 GB RAM using CVXPY with the Clarabel solver, median solve time per step is $< 1$ s for $N \leq 7$. Accordingly, considering that the optimisation problem is solved every time step, and here $\Delta_t = 30$ min, computational time is not an issue.

## 3 Estimation of the input sequences

### 3.1 The long-range carbon intensity forecast, $\hat{C}_{gen}[l]$

The objective (2) minimizes the predicted net $CO_2$, $\hat{J}_0$, at arbitrary datum, $l = 0$, over the subsequent $N$-day forecast window, in which the predicted carbon intensity is $\hat{C}_{gen}[l]$, $l \in \{1, \ldots, c \times N\}$ (1). The minimization is repeated every $\Delta_t$ hours, by sliding the forecast window forward in time, i.e., $l \leftarrow l + 1$. Note that NESO makes available the *actual* half-hourly carbon intensity sequence, $C_{gen}[l]$, as well as its half-hourly, short-range forecast up to $N = 2$ days ahead. However, only the one-step-ahead NESO prediction is actually stored. We will specifically denote this NESO-stored one-step-ahead prediction of carbon intensity at time interval, $l$, by $\hat{C}_{gen}^{(1)}[l]$ (Fig 1). Therefore, our requirement for the carbon minimization objective (2) is to synthesize the $l$-step-ahead carbon intensity forecast sequence, $\hat{C}_{gen}[l]$, over the long-range forecast window, in a manner which captures its increasing variance with $l$, as seen in the example in Fig 1. We highlight, that our objective in this section is not to propose a carbon intensity forecasting model, but rather to synthesize realistic long-range forecasts from NESO one-step-ahead data combined with published error growth statistics. [21]

To summarize, NESO's publicly available data [25] provides the following sequences at any arbitrary datum ($l = 0$):

- $C_{gen}[l]$, the actual half-hourly (average) carbon intensity sequence for the UK national grid, indexed by $l \in \{1, \ldots, c \times N\}$;

- $\hat{C}_{gen}^{(1)}[l]$, the associated *one-step-ahead* forecast of the carbon intensity in time interval, $l$.

#### 3.1.1 Relative error of the $l$-step-ahead forecast. This is defined as follows:

$$\hat{\epsilon}[l] = \frac{\hat{C}_{gen}[l] - C_{gen}[l]}{C_{gen}[l]}, \tag{8}$$

Hence:

$$\hat{C}_{gen}[l] = C_{gen}[l]\left(1 + \hat{\epsilon}[l]\right). \tag{9}$$

Hence, if we can estimate—from available data—the $l$-step-ahead forecasting error, $\hat{\epsilon}[l]$, we can retrospectively synthesize the associated multi-step-ahead (i.e., long-range) noisy forecast—required in our objective (2)—from the NESO-stored sequence of actual carbon intensities, $C_{gen}[l]$.

#### 3.1.2 Transferring CarbonCast [21] side-information for linear scaling of the forecast error. The mean absolute percentage error (MAPE) statistics for carbon intensity forecasts up to four days have been published in [21], using those authors' own CarbonCast forecasting technique along with two alternative forecasting techniques. As explained below, these statistics support a hypothesis of linear growth in the expected MAPE of these long-range forecasts. Under this hypothesis, we therefore write

$$|\hat{\epsilon}[l]| = |\hat{\epsilon}[1]| \times \left(1 + \hat{\lambda} \times (l - 1)\right), \tag{10}$$

where $\hat{\lambda}$ is the estimated constant rate of growth of the forecast MAPE.

Recall that the one-step-ahead relative prediction error, $\hat{\epsilon}[1]$, can be computed directly from the published and stored NESO sequences, $C_{gen}[1]$ and $\hat{C}_{gen}^{(1)}[1]$, via (8). The empirical distribution of the NESO published one-step-ahead relative prediction errors is illustrated in Fig 2.

Obviously, the absolute error sequence (10) suppresses the sign (i.e., polarity) of each $\hat{\epsilon}[l]$, which we denote by $a[l] \in \{+1, -1\}$. We recover these in our simulated forecasts via iid Bernoulli trials of probability $\frac{1}{2}$. Then:

$$\hat{\epsilon}[l] = a[l] \times |\hat{\epsilon}[l]|. \tag{11}$$

Inserting (11) and (10) into (9), we obtain a formula for synthesizing the long-range carbon intensity forecast sequence in terms of the published NESO data, with knowledge transfer (into $\hat{\lambda}$) from the CarbonCast data [21]:

$$\hat{C}_{gen}[l] = C_{gen}[l] \times \left[ 1 + a[l] \times |\hat{\epsilon}[1]| \times \left( 1 + \hat{\lambda} \times (l - 1) \right) \right]. \tag{12}$$

**Remark: Linear error growth** While our assumption of linear error growth provides a simple and conservative approximation, actual error dynamics may deviate, e.g., exhibiting concavity or saturation at long horizons. Future work may refine this model by exploiting richer UK-specific forecast datasets or by adopting a non-parametric modelling technique.

**3.1.3 Computation of $\hat{\lambda}$.** We compute the error growth rate per half-hour interval to be $\hat{\lambda} \approx 9.97 \times 10^{-3}$, based on the published forecasts in [21], in which the MAPEs exhibit an approximately linear (albeit slightly concave) increase up to a horizon of four days, and across multiple regions, excluding the UK. UK-specific data are not reported, and so we estimate them via the average growth rate across the actually reported regions. Calibration with UK-specific data may refine this approach in the future. The linear model adopted here is a reasonable simplification which leads to a conservative estimate, since it overestimates long-range forecast errors. Fig 1 shows a typical comparison between our synthetic forecast, $\hat{C}_{gen}[l]$ (12), and the *actual* stored NESO carbon intensity sequence, $C_{gen}[l]$. In addition, Fig 3 illustrates the mean absolute percentage error (MAPE) of the synthesized long-range carbon intensity forecast sequence, $\hat{C}_{gen}[l]$ for the year 2022. As can be observed, MAPE grows linearly with forecast horizon, in line with the assumption in 10. As $l$ increases, errors accumulate in the synthetic forecast, approximating the growing uncertainty over longer forecast horizons. Overall, the method described here provides a practical way to evaluate our charging strategy subject to the increasing forecast error exhibited by the published data.

**Remark: Regional Differences in $\hat{\lambda}$** Our calibration of $\hat{\lambda}$ relies on region-averaged values from [21], since UK-specific data were not available. It is important to clarify, that the value of the error growth rate per half-hour may significantly change from one area to another area. As we have already mentioned, the carbon intensity largely depends on

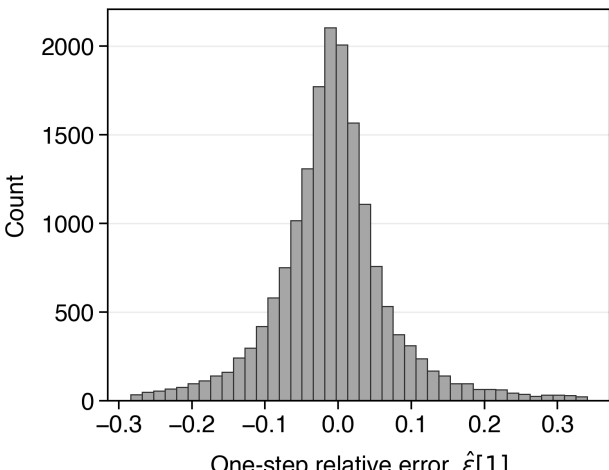

**Fig 2. Empirical distribution of relative one-step ahead forecast error, $\epsilon[1]$, observed in the NESO published forecasts (2022) [25].**

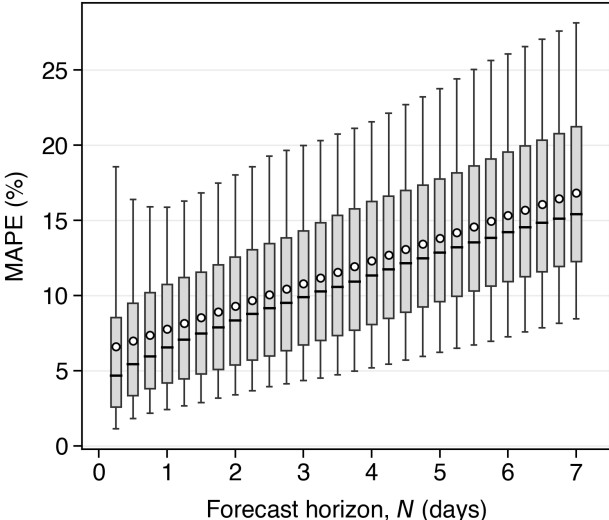

**Fig 3. Mean absolute percentage error (MAPE) of the synthesized long-range carbon intensity forecast sequence, $C_{gen}[l]$.**

how much energy has been generated from renewable sources, and it may be simpler to predict when energy is mainly generated from solar plants, for instance, while it may be harder to predict when it is mainly generated from wind farms, where generation forecasts are less accurate. Future work may refine this model by exploiting richer UK-specific long-range forecast datasets.

### 3.2 Expected start and end times, $\hat{k}_{s,b}$ and $\hat{k}_{s,e}$, of charging sessions

The start and end times of the charging sessions in (2)—quantized and indexed by $l = \hat{k}_{s,b}$ and $l = \hat{k}_{s,e}$ for the $s$th session, $s \in \{1, \ldots, N\}$—are chosen in order to represent a typical daily commuting pattern, where plugging in the EV overnight is part of the daily routine, in line with the 'routine charging schedules' considered in [15]. Plug-in and -out times and their standard deviations are modelled based on average values from a data set of domestic charging events [26]. Accordingly, the simulated times are modelled via normal distributions with a one-hour standard deviation, centred around 18:00 (start) and 09:00 (end). We then choose the $s$-invariant start time ($\hat{k}_{s,b} \equiv \hat{k}_b$) and end time ($\hat{k}_{s,e} \equiv \hat{k}_e$) as the 98th percentile (i.e., 20:03) and 2nd percentile (i.e., 06:57), respectively. These choices lead to a conservative design (i.e., over-design) of the charging power sequence, $P[s,k]$, in (2), ensuring that the EV is sufficiently charged with high probability, even if it is plugged in unusually late, and plugged out unusually early.

### 3.3 Expected daily energy consumption, $\hat{E}_s$

The daily estimated energy consumption values, $\hat{E}_s$, for the $N$ charging sessions (4), are also computed $s$-invariantly via a stationary normal model:

$$\hat{E}_s \overset{iid}{\sim} \mathcal{N}(5.8, 2.67^2).$$

Here, the mean and standard deviation are informed by the Electric Chargepoint Analysis dataset of the Department for Transport [26]. The actual $s$-invariant value, $\hat{E}_s \equiv \hat{E}$, inserted into (4), is again chosen conservatively, as the 98th percentile value ($\hat{E} \approx 11.28$ kWh), to ensure the over-design of the charging power sequence, $P[s,k]$, in (2). In a practical deployment, precautionary next-day energy requirements need not be manually entered by users: they can be learned from

historical charging/driving data (e.g., high-percentile recent daily energy use, possibly conditioned on weekday/season) and presented as a default that users can override when anticipating unusually long trips. We note here, that better results could be obtained if less conservative estimates would be provided to the charging system.

## 4  Simulated charging scenarios

Here, we evaluate the effectiveness of the proposed MPC strategy (see Section 2) for minimizing carbon emissions in the charging processes, through a number of different numerical simulations. All case studies are simulated for a period of one year (2022) based on the carbon intensity of the UK national electricity grid [25]. EV and charge point parameters are specified to resemble typical mid-range EVs and domestic charging stations. The battery capacity is set at 50kWh and the maximum grid-side charging power at 10kW with simplified charging efficiency $\eta = 1$. Because we use full-year, half-hourly historical UK grid carbon intensity data for 2022, the input signal embeds real renewable variability (including solar and wind intermittency) and its interaction with demand and conventional generation. To account for uncertainty in daily driving demand and charging availability, we run 10 independent Monte Carlo simulations, each with randomized sequences of daily energy requirements and plug-in times, drawn from the statistical distributions described in Section 2. Reported results present the mean and standard deviation across these Monte Carlo runs.

### 4.1  Annual emission reductions with N-day MPC charging

Table 2 summarizes the outcome of the simulations for a whole year across a range of forecast horizons $N \in \{1, \ldots, 7\}$ days ahead. As can be noticed, the average carbon intensity of energy charged by the EV, $C_{EV}$ (gCO$_2$/kWh), initially falls steeply from the uncontrolled scenario to MPC with $N=1$ and $N=2$, and continues to improve up to $N=4$, where it reaches its minimum at 126.73 gCO$_2$/kWh — a 37.41% reduction compared to uncontrolled charging. The marginal relative improvement of multi-session optimization ($N>1$) over single session optimization ($N=1$) is also maximised at $N=4$, at 22.89%. Interestingly, increasing the future horizon of time beyond 4 days does not seem to provide further improvements, but instead may lead to higher $C_{EV}$, since the additional temporal flexibility of the longer horizon is counterbalanced by the increased inaccuracy of the carbon intensity forecast. The diminishing benefit of extending the forecast horizon beyond four days is further illustrated in Fig 4, where the mean and standard deviation of $C_{EV}$ across 10 Monte Carlo runs are shown. As can be seen, $C_{EV}$ falls sharply from $N=1$ to $N=2$, plateaus around $N=4$, and rises slightly for $N \geq 5$.

  4.1.1  **Seasonality of emission reductions.**  In this section we evaluate the impact of seasonality patterns on our results in our case study. Fig 5 shows the average carbon intensity of EVs both in the uncontrolled case, and when the proposed smart charging methodology is implemented, for different future horizons of time. We consider $N=1,2,4$, as no particularly benefits were observed for longer horizons as described in Section 4.1. We observe first that in our case study

Table 2.  **Aggregate simulation results (2022, UK National Grid). Mean ± standard deviation of average carbon intensity of energy charged by the EV, $C_{EV}$ (gCO$_2$/kWh) and percentage improvements for different forecast horizons N.**

| N | $C_{EV}$ (gCO$_2$/kWh) | Impr. vs. Unctrl (%) | Impr. vs. N=1 (%) |
|---|---|---|---|
| 0 | 202.49 ± 2.23 | — | — |
| 1 | 164.35 ± 2.65 | 18.84 ± 0.52 | — |
| 2 | 137.18 ± 2.48 | 32.26 ± 0.54 | 16.53 ± 0.51 |
| 3 | 127.06 ± 1.81 | 37.25 ± 0.37 | 22.69 ± 0.48 |
| 4 | 126.73 ± 2.06 | 37.41 ± 0.41 | 22.89 ± 0.40 |
| 5 | 126.82 ± 2.00 | 37.37 ± 0.52 | 22.83 ± 0.57 |
| 6 | 128.15 ± 1.75 | 36.71 ± 0.35 | 22.02 ± 0.60 |
| 7 | 129.74 ± 1.96 | 35.93 ± 0.41 | 21.06 ± 0.36 |

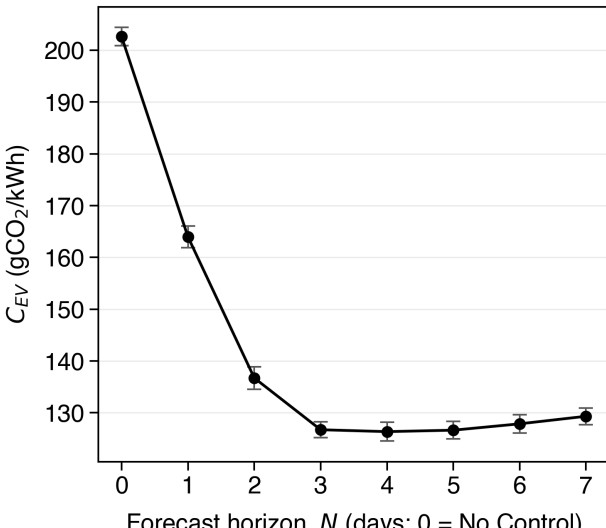

**Fig 4. Sensitivity of $C_{EV}$ to forecast horizon $N$.** The error bars show one standard deviation around the mean. The sensitivity analysis suggests that $C_{EV}$ plateaus for approximately $N > 3$.

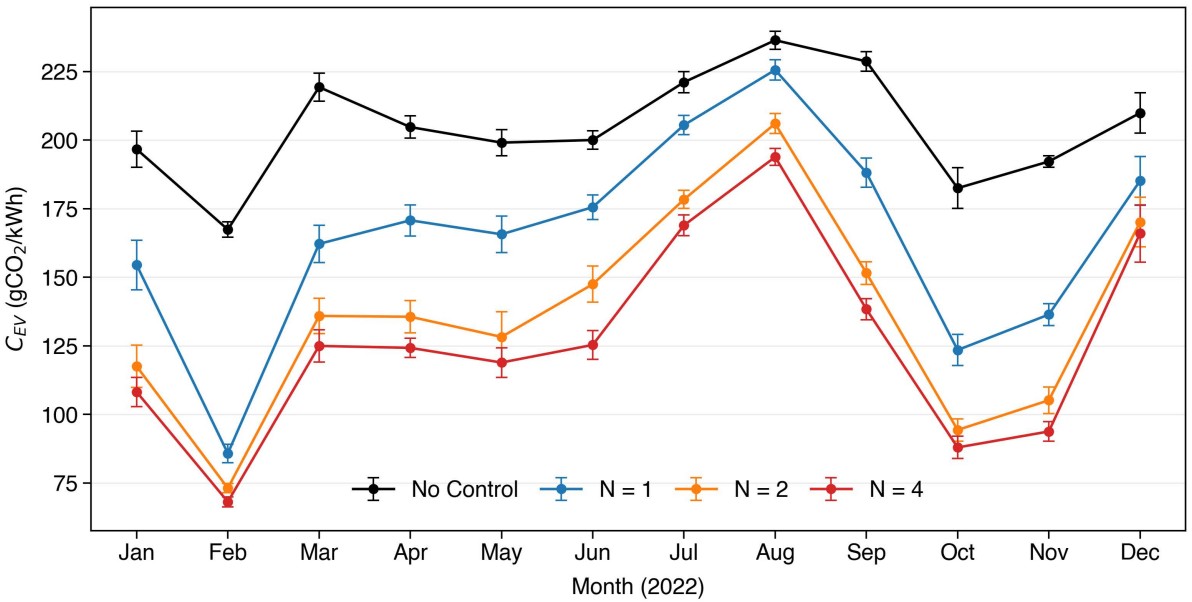

**Fig 5. Average carbon intensity of energy charged by the EV, $C_{EV}$ for each month of the year 2022.**

$C_{EV}$ is rather constant during the whole year, with an average value of about 200 gCO$_2$/kWh. In addition, we show that the proposed methodology consistently outperforms the uncontrolled charging, on average, every month of the year. Likewise, the same trend of the future horizon of prediction is confirmed every month.

**4.1.2 Mechanism of emission reductions.** To explain the mechanism behind the emission reductions achieved by N-day MPC charging, Figs 6–8 compare uncontrolled charging (a) with two selected MPC strategies at forecast horizons

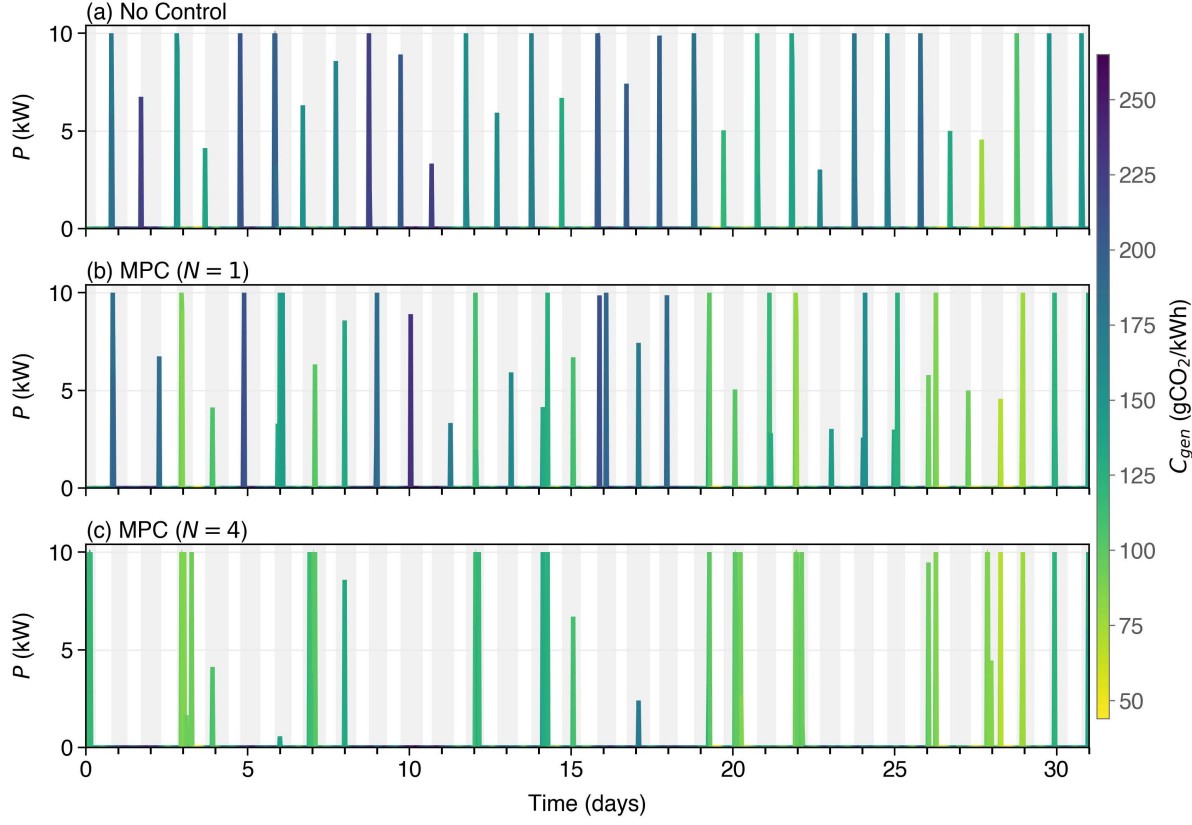

**Fig 6. Charging Power, P (kW), during a period of one month for three different Charging Strategies: (a)** uncontrolled plug-and-charge strategy; **(b)** MPC (N = 1) charges the EV every night, but at the expected minimum carbon intensity during that night; **(c)** MPC (N = 4) can delay charging during nights where $C_{gen}$ is high, resulting in fewer, but longer, charges during periods of low $C_{gen}$.

of N = 1 (b) and N = 4 (c). The gray background rectangles identify night charging sessions, while white regions indicate daytime hours when the EV is not plugged in.

Fig 6 highlights the power output of the domestic charger during a typical period of one month for the three charging strategies. The power profile is coloured to represent the carbon emissions signal, $C_{gen}$, during power delivery, with dark blue hue indicating a large value of $C_{gen}$ and yellow indicating low values. The uncontrolled strategy (a) charges the EV every night, as soon as it is connected to the plug. The amount of charging depends on how much energy has been depleted during the past day. Single session optimization (b) also charges the EV every night, but it may delay the time at which the charging events start, so as to select the time of lowest $C_{gen}$ during the night. Finally, multi-session optimization (c) reduces the number of nights when charging events take place. It selects those nights when carbon emissions, $C_{gen}$, are expected to be low, avoiding those when $C_{gen}$ is expected to be relatively high. In order to compensate for fewer charging events, each one takes more time (so that lines appear thicker in the figure).

Fig 7 compares the same strategies also showing the evolution of the SOC, as a function of the $C_{gen}$ signal (in black), to emphasize that every morning a minimum SOC is guaranteed. Again, a period of one week is chosen for clarity. As can be seen, the uncontrolled strategy (blue) immediately charges the EV at the beginning of the gray rectangles (night charging sessions), and the morning SOC always reaches the upper SOC limit. Similarly, the single session MPC strategy (orange) shifts the charging process during the night. The MPC strategy with a longer future horizon of time (N = 4, green) prefers not to charge the EV during some of the nights, if the carbon intensity is not convenient, but in this case as well,

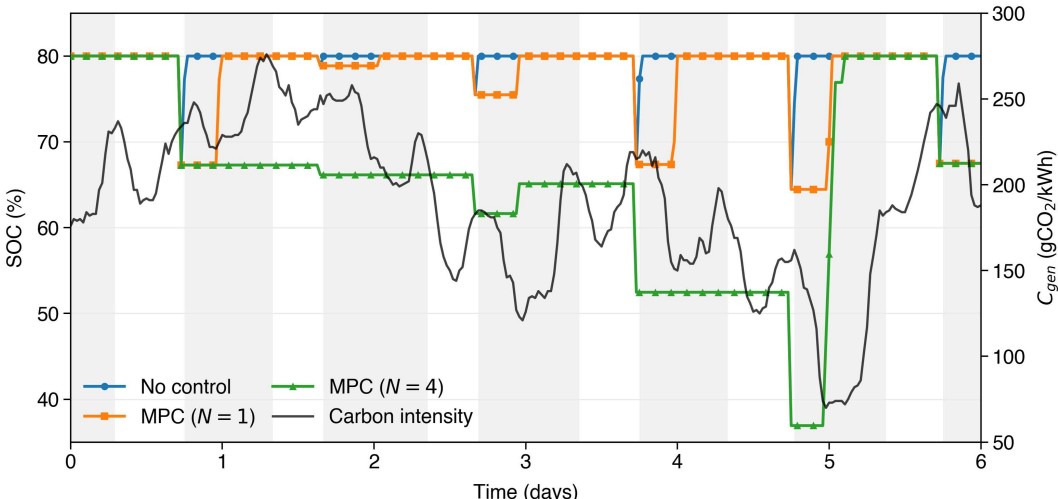

**Fig 7. State of Charge (SOC) profiles over six simulation days for three charging strategies: uncontrolled (blue), MPC with a 1-day horizon (orange), and MPC with a 4-day horizon (green).** The black line shows the carbon intensity signal ($C_{gen}$). Gray background indicates night charging windows, during which the EV is plugged in; The figure highlights how predictive strategies shift charging to periods of lower carbon intensity.

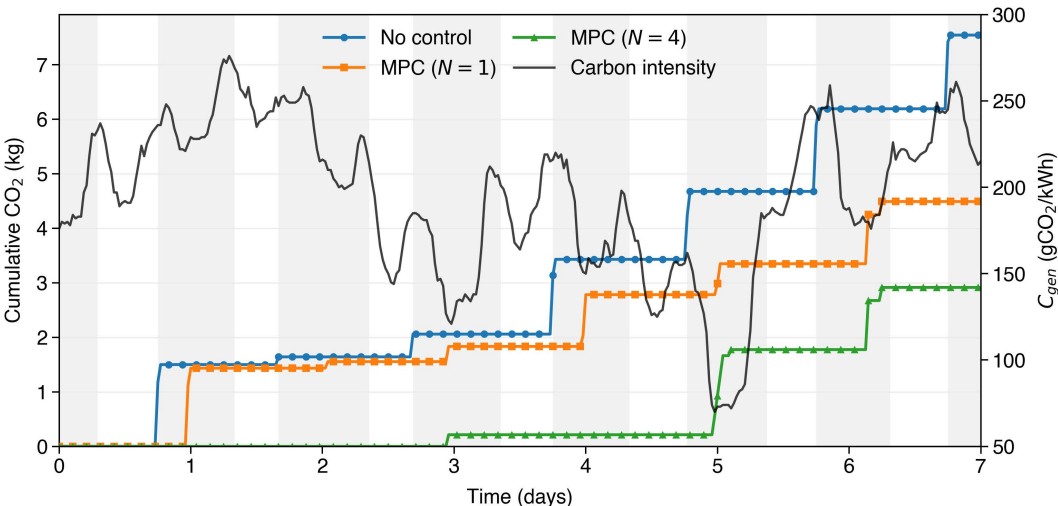

**Fig 8. Cumulative $CO_2$ emissions over six simulation days for the same three charging strategies.** Emissions increase stepwise with each charging event. MPC strategies result in significantly lower cumulative emissions.

the morning SOC (i.e., at the beginning of the white rectangles) is always sufficient to meet next days expected energy demand.

Finally, Fig 8 allows one to evaluate the cumulative advantage of one strategy over the others in terms of reduced carbon emissions. In particular, it depicts the cumulative carbon intensity of the three strategies during the same one week period, and it increases to a new value every time a charging event is concluded. The MPC strategies obtain significantly reduced amounts of $CO_2$, and with a forecast horizon of N=4 days, the amount is about 50% of the amount of the uncontrolled case.

## 4.2 Effect of user flexibility on emission reduction

The carbon intensity that is achieved when employing such smart charging approaches ultimately depends on the driving and charging patterns of the user. For example, if the EV is plugged in for a long duration, or at the right times, the controller is more likely able to charge during periods of low carbon intensity. On the other hand, if the EV is connected infrequently, such low-carbon periods may be missed, resulting in a higher average carbon intensity. Similarly, if the user consistently uses little energy for driving, then the smart charging algorithm has a greater flexibility to delay charging until convenient periods of low carbon intensity are met. On the other hand, if the battery is depleted every day, then there is no such flexibility, and the battery may have to be charged for the whole night. Fig 9 illustrates how the average carbon intensity of energy charged by an EV using multi-session predictive control (N = 4) varies, depending on both the plug-in time frame, as well as the daily energy demand of the driver. These results are based on a single simulation month (January 2023) with UK National grid carbon intensity. The grey bars illustrate the daily plug-in periods, with plug-in durations ranging between 20 and 4 hours, decreasing in steps of two hours. Overnight charging scenarios are represented on the left side of the figure, and daytime charging scenarios on the right side. The four lines show the average carbon intensity achieved based on daily energy demands of 5,10,20, and 30 kWh—approximately equivalent to a daily driving distance of 25, 50, 100, and 150 km. We can see, that plugging in for shorter periods generally leads to higher carbon intensity for all energy demands—slightly higher (≈8%) for overnight scenarios and significantly higher (≈25%) for daytime charging scenarios. With the exception of the first two daytime scenarios (20 and 18 hours total plug-in duration), charging during the day leads to higher carbon intensity than charging over night, even if plug-in times are longer. The figure further shows that EVs with small daily energy demands (e.g., 5kWh) can charge less carbon intensive energy on average. In comparison, the energy charged by drivers that require 30kWh of energy every day is on average 27.3% more carbon intensive.

Overall, these findings highlight that individual driving and charging patterns influence the carbon intensity of EV charging. A natural next step is thus to explore how to incentivize "virtuous" behaviours, i.e., reduced usage of the EV or leaving it plugged in as much as possible, that may lead to reduced charging-related emissions. One way to do this could be through emission markets or pooling mechanisms, which create financial incentives for reducing EV-related emissions

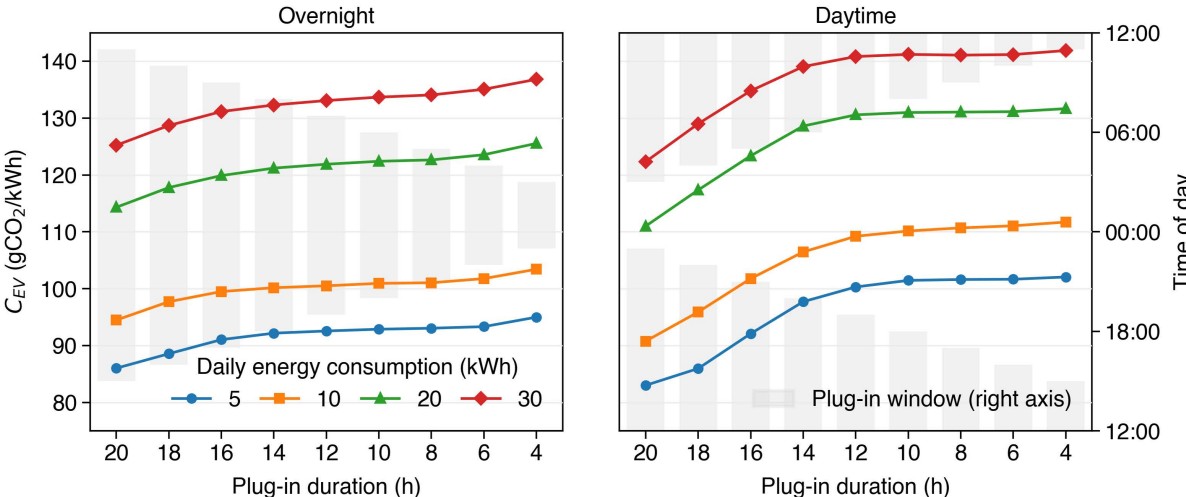

**Fig 9. Impact of user flexibility on average carbon intensity ($C_{EV}$) during one month of simulated EV charging (January 2023, UK national carbon intensity data).** Each gray bar represents a different daily plug-in time window, ranging from 20 to 4 hours. Four lines correspond to daily energy demands of 5, 10, 20, and 30 kWh. Shorter plug-in durations and higher energy demands both increase average carbon intensity. Daytime charging scenarios (right side) typically result in significantly higher emissions than overnight charging at comparable plug-in durations.

below a threshold. A similar scheme is well-established in the EU to reduce $CO_2$ emissions from passenger cars and light commercial vehicles [27]. Similar schemes could be adopted for charging EVs as well, with the goal to incentivize EV users to plug in overnight and for longer periods, allowing them to benefit financially from selling emission reduction credits to users with less flexibility. The design and empirical testing of such schemes presents a promising direction for future research.

### 4.3 Regional variation in emission reduction potential

Regional factors also shape the effectiveness of smart charging. As highlighted in the introduction, the GWP of power generation varies significantly between countries and regions. In this section, we simulate the potential of MPC-based smart charging with prediction horizons $N = 1,2,4$ days in 14 different UK regions. UK Regional data on the GWP of electricity in 2023 for this simulation is published by the National Energy System Operator in half-hourly timesteps [25]. Notably, the GWP of electricity shows significant regional variation and is much lower in northern regions, such as Scotland, North England and North Wales. On average, each unit of electricity in South Wales emits more than three times more carbon emissions when compared to these northern regions. Fig 10 shows the average carbon intensity achieved by uncontrolled charging versus MPC charging (Section 2) during a simulation of one month (January 2023). It is evident, that (1) Smart charging reduces the carbon intensity of EV charging effectively across all regions of the UK, with relative improvements ranging between 16 and 83.9% depending on region and prediction horizon. (2) Longer prediction horizons (e.g., N = 4) generally lead to higher reductions in GWP, with the average percentage reduction for prediction horizons of N = 1,2,4 being 39.2%, 55.5%, and 59.2% respectively. (3) The absolute reduction in GWP achieved through smart charging is particularly relevant in those regions with a higher average GWP of electricity. For example, the absolute reduction in average GWP of energy charged in South Wales using Smart Charging with 4 day prediction horizon is 175 $gCO_2$/kWh, while the

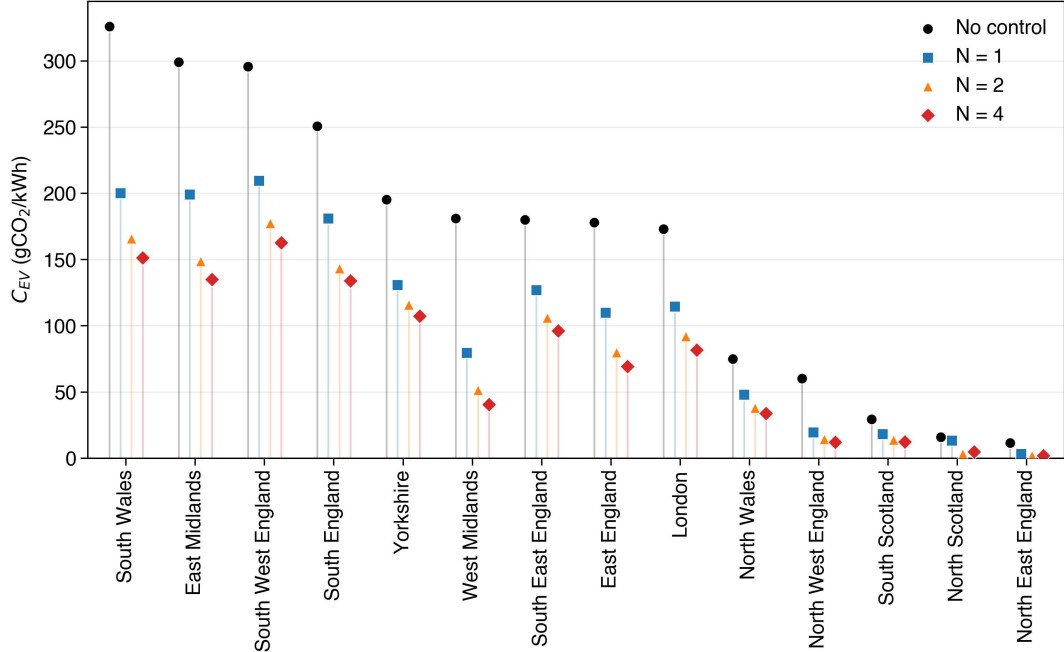

**Fig 10. Regional variation in average carbon intensity ($C_{EV}$) of EV charging across 14 UK regions during January 2023.** Results are shown for uncontrolled charging and three predictive MPC strategies with forecast horizons N = 1, 2, 4 days. Smart charging reduces emissions in all regions, with relative reductions ranging from 16% to 83.9%. Longer prediction horizons yield larger reductions across regions.

respective absolute reduction in North East England is only 9.8 $gCO_2$/kWh. Here, even uncontrolled charging is relatively clean at 11.7 $gCO_2$/kWh, leaving little potential for improvement. Finally, (4) the relative reduction achieved through smart charging against uncontrolled charging varies between regions, suggesting that the individual carbon intensity profile of a region determines the relative emission reduction potential for smart charging. However, in all cases, independently from the specific carbon intensity of power generation, the MPC scheme appears to significantly reduce the emissions of the uncontrolled strategy.

## 5 Conclusions

This paper presents a multi-session MPC strategy for overnight EV charging that leverages forecasts of grid carbon intensity to reduce charging-related $CO_2$ emissions, while constraining the SOC to levels aligned with reduced battery degradation. Results show that scheduling charges over multiple sessions (up to one week ahead) outperforms session-by-session MPC and uncontrolled "plug-and-charge" approaches, enabling emission reductions of up to 37% compared to uncontrolled charging and up to 23% compared to an otherwise identical single night optimization. Carbon emission savings are enabled by scheduling charging power during multi-day minima of grid carbon intensity and by exploiting the flexibility to delay charging of energy that is not required for next day's driving.

Our simulations confirm, that carbon emission reductions depend on the energy need and charging behaviour of the user, as well as the GWP profile of the local power grid. Longer and well-timed plug-in durations as well as modest daily energy requirements contribute to enhanced emission reductions, and significant absolute savings can be achieved in regions with high average GWP of electricity. Accordingly, the proposed strategy may be simply adopted to accelerate the decarbonization process of the electric mobility sector, without requiring significant infrastructure upgrades or new technology developments. Our work shows that significant improvements to the carbon intensity of EV charging processes can be achieved with simple domestic smart plugs. Beyond significant direct emission reductions from charging, the proposed strategy may also lower indirect emissions by maintaining SOC at levels associated with slower battery degradation, thus slowing battery replacement. The utilization of a fully coupled State-of-Health (SOH) – carbon model (e.g., using more sophisticated linear and nonlinear SOH models [2]) is an interesting extension of our work, also to verify the expected increased battery lifetime. Although the case study is scoped to domestic wired charging using smart plugs, the proposed multi-session MPC framework is hardware-agnostic and can, in principle, be interfaced with other charging technologies, including wireless charging systems, provided that charging power (or its time-averaged equivalent) can be actuated and an availability window is known. Exploring such integrations and their interoperability constraints is a natural direction for future work. A limitation of this study is the reliance on average emission factors to estimate carbon savings. While these provide a system-wide, long-term perspective appropriate for our multi-session analysis, they do not capture marginal system responses to incremental charging demand [16,17]. This limitation is even more pronounced when considering fleet-scale applications: if thousands of EVs synchronously follow the same strategy, then we cannot tacitly assume any more that the charging behaviour of EVs is decoupled from the carbon intensity curves. Conversely, it may become necessary to design an aggregator entity to coordinate the charging of the fleet of EVs, and possibly to supervise emission trades between EV participants. Exploring this possibility – together with the realization of a device for achieving the proposed smart charging procedure – are interesting extensions of our work.

## Author contributions

**Conceptualization:** Felix Wieberneit, Emanuele Crisostomi, Robert Shorten.

**Data curation:** Felix Wieberneit.

**Funding acquisition:** Robert Shorten.

**Investigation:** Felix Wieberneit, Emanuele Crisostomi, Anthony Quinn, Homayoun Hamedmoghadam, Robert Shorten.

**Methodology:** Felix Wieberneit, Emanuele Crisostomi, Anthony Quinn.

**Software:** Felix Wieberneit.

**Supervision:** Robert Shorten.

**Validation:** Emanuele Crisostomi, Anthony Quinn, Homayoun Hamedmoghadam.

**Visualization:** Felix Wieberneit.

**Writing – original draft:** Felix Wieberneit.

**Writing – review & editing:** Emanuele Crisostomi, Anthony Quinn, Homayoun Hamedmoghadam.

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
