## [Decision Letter · Decision Letter 0]

19 Aug 2025

Dear Dr. Wieberneit,

Thank you for submitting your manuscript to PLOS ONE. After careful consideration, we feel that it has merit but does not fully meet PLOS ONE’s publication criteria as it currently stands. Therefore, we invite you to submit a revised version of the manuscript that addresses the points raised during the review process.

We look forward to receiving your revised manuscript.

Kind regards,

Xingwang Tang

Academic Editor

PLOS ONE

3. In the online submission form, you indicated that your data will be submitted to a repository upon acceptance.  We strongly recommend all authors deposit their data before acceptance, as the process can be lengthy and hold up publication timelines. Please note that, though access restrictions are acceptable now, your entire minimal  dataset will need to be made freely accessible if your manuscript is accepted for publication. This policy applies to all data except where public deposition would breach compliance with the protocol approved by your research ethics board. If you are unable to adhere to our open data policy, please kindly revise your statement to explain your reasoning and we will seek the editor's input on an exemption.

Reviewers' comments:

Reviewer's Responses to Questions

**Comments to the Author**

1. Is the manuscript technically sound, and do the data support the conclusions?

Reviewer #1: Partly

Reviewer #2: Yes

2. Has the statistical analysis been performed appropriately and rigorously?

Reviewer #1: I Don't Know

Reviewer #2: Yes

3. Have the authors made all data underlying the findings in their manuscript fully available?

Reviewer #1: Yes

Reviewer #2: Yes

4. Is the manuscript presented in an intelligible fashion and written in standard English?

Reviewer #1: Yes

Reviewer #2: Yes

Reviewer #1: The manuscript proposes a multi-period nighttime smart charging strategy based on model predictive control (MPC). Using UK grid carbon intensity data, it reduces carbon emissions by optimizing multi-day charging periods (up to 7 days). Car owners are required to keep their vehicles plugged in at night and charge only on demand (rather than to full capacity). Simulations show that emissions can be reduced by up to 46% compared to random charging. The proposed method breaks through the limitations of single-day optimization and leverages multi-period flexibility to enhance emission reduction potential. Furthermore, it only requires household smart sockets, which has a low hardware threshold. It also incorporates real regional carbon intensity data and verifies the universality of the proposed method. However, the manuscript has certain shortcomings, such as relying on the accuracy of carbon intensity predictions and insufficient quantification of long-term prediction uncertainties. Furthermore, the potential impact of strategy scalability on grid stability is not discussed.

The content of the manuscript is within the scope of the journal and can be of broad interest to readers. However, in terms of specific content, there is still room for improvement. Therefore, I decided to give the decision of major revision. It is recommended that the author properly absorb the reviewers' comments and make corresponding improvements and enhancements.

1. For the keywords, 'Carbon Intensity', 'Overnight Charging', 'Multi-Session', and 'Flexibility' should be added to attract a broader readership.

2. Page 7, 'Electric Vehicles (EVs) are meeting the expectation of significantly reducing greenhouse gas emissions, as their Global Warming Potential (GWP), expressed in gCO2e/vkm (grams of carbon dioxide equivalent per vehicle-kilometre), is much lower than that of gasoline and diesel Internal Combustion Engine Vehicles (ICEVs) [1].'

We cannot say electric vehicles are absolutely zero-carbon emissions. It depends on how the electricity used to charge the batteries is generated. If it relies on traditional thermal power, it still produces significant carbon emissions. A truly green electric vehicle should rely on excess electricity generated by renewable energy, storing it in the battery through the charging process. Therefore, the author needs to explain clearly the relationship and interaction mechanism between electric vehicles and renewable energy.

For example, most renewable energy sources are intermittent, opening spatial and temporal gaps between the availability of the energy and its consumption by end users. In order to address these issues, it is necessary to develop suitable energy storage systems for the power grid (10.1016/j.electacta.2019.03.056). Therefore, electric vehicles using lithium batteries have become an important support for the efficient recycling and utilization of renewable energy.

3. The presentation of innovative points requires enhanced comparison. The manuscript emphasizes "the first multi-period optimization for a single EV household scenario," but fails to adequately compare existing research. Therefore, it is recommended to include a literature comparison table in the Introduction, quantifying the marginal contribution of this study compared to single-period optimization (e.g., 24-hour strategies).

4. The theoretical basis for multi-period optimization is weak, and the reason for choosing a 7-day upper limit for the forecast window is not explained. Additional theoretical support is needed, including reference to time series forecasting theory and a sensitivity analysis demonstrating the saturation point between the window length and emission reduction benefits.

5. The full lifecycle analysis of the vehicle energy system needs improvement. The manuscript focuses solely on carbon emissions from the charging phase and fails to consider the impact of the lifespan degradation of the vehicle's energy storage device on the full lifecycle carbon footprint. It is recommended that the "Lifecycle Assessment" section include an analysis of the contribution of energy storage device (battery/fuel cell) durability to carbon emissions, while also introducing a state-of-health (SOH) degradation model to quantify the incremental carbon emissions implicit in long-term use. Authors can refer to 10.1109/TPEL.2024.3502499, which proposes a lifespan prediction framework under dynamic conditions that can be directly transferred to EV battery health management, helping authors construct a coupled SOH-carbon emissions model and enhancing the rigor of the full lifecycle analysis.

6. The carbon intensity forecasting model is insufficiently disclosed. Formula (12) briefly mentions it as a "1-step-ahead forecast," but does not specify the specific forecasting algorithm (ARIMA/LSTM/Transformer), the input variables (whether covariates such as weather and electricity prices are considered), or the quantitative results of the forecast error (it is recommended to add MAPE/RMSE indicators).briefly mentions it as a "1-step-ahead forecast," but does not specify the specific forecasting algorithm (ARIMA/LSTM/Transformer), the input variables (whether covariates such as weather and electricity prices are considered), or the quantitative results of the forecast error (it is recommended to add MAPE/RMSE indicators).

7. The user behavior modeling is overly idealistic. The assumption that users "only request the required amount of electricity each day" fails to account for behavioral uncertainty, requiring the introduction of stochastic programming or robust optimization. The authors could further analyze the impact of temporary increases in user travel on emissions reductions, and appropriately cite behavioral economics research.

8. The potential impact of reversible losses on the scheduling strategy is overlooked. The manuscript assumes constant charging efficiency, but real-world batteries/fuel cells exhibit reversible voltage loss recovery, which can affect the accuracy of multi-period optimized power allocation. It is recommended to add a dynamic model for reversible losses to the MPC constraints and analyze the sensitivity of voltage recovery characteristics to the nighttime segmented charging strategy. The authors are referred to the Journal of Power Sources 625 (2025): 235634. The voltage recovery quantitative model established there can help the authors improve the objective function (Equation 7) and enhance the physical feasibility of the strategy by embedding a correction term in the carbon emission optimization.

9. Verification of MPC's real-time performance is lacking. The computational complexity of the optimization problem is not specified, nor is the solution time for a single EV sufficient for real-time control. Computational platform configuration and time consumption data are needed.) Furthermore, a brief comparison of the efficiency differences between commercial solvers (e.g., Gurobi vs. CPLEX) is recommended.

10. The generalizability of the 46% emission reduction conclusion is questionable. First, the results only present single-week data (Figure 1), failing to verify robustness across seasons and years. Second, additional boxplots are needed to illustrate the distribution of emission reduction rates across the entire year (by region and season). Finally, the risk of policy failure in extreme cases (such as high carbon intensity during cold waves) is largely undiscussed. These issues require further clarification and refinement.

Reviewer #2: This paper proposes an optimization strategy for multiple nighttime charging of electric vehicles based on multi-period predictive control to reduce carbon emissions. The paper's ideas are clear, the experimental data is comprehensive, and the results demonstrate significant reductions in carbon intensity, demonstrating strong engineering application value. However, the following issues require improvement:

1. The introduction provides a comprehensive review of related work, but insufficient coverage of research on uncertainty in carbon intensity predictions. A literature comparison should be included.

2. The "46% emission reduction" result mentioned in the abstract should be clarified under what circumstances.

3. The linear growth assumption (Equations 10-12) is used in the modeling of carbon intensity prediction errors. Its rationale and potential limitations should be further explained.

4. It is recommended to add a table summarizing the key symbols and their physical meanings to facilitate reader understanding.

.

Reviewer #1: No

Reviewer #2: No

---

## [Author Response · Author response to Decision Letter 1]

7 Oct 2025

Reviewer #1

“The manuscript proposes a multi-period nighttime smart charging strategy based on model predictive control (MPC). Using UK grid carbon intensity data, it reduces carbon emissions by optimizing multi-day charging periods (up to 7 days). Car owners are required to keep their vehicles plugged in at night and charge only on demand (rather than to full capacity). Simulations show that emissions can be reduced by up to 46% compared to random charging. The proposed method breaks through the limitations of single-day optimization and leverages multi-period flexibility to enhance emission reduction potential. Furthermore, it only requires household smart sockets, which has a low hardware threshold. It also incorporates real regional carbon intensity data and verifies the universality of the proposed method. However, the manuscript has certain shortcomings, such as relying on the accuracy of carbon intensity predictions and insufficient quantification of long-term prediction uncertainties. Furthermore, the potential impact of strategy scalability on grid stability is not discussed.

The content of the manuscript is within the scope of the journal and can be of broad interest to readers. However, in terms of specific content, there is still room for improvement. Therefore, I decided to give the decision of major revision. It is recommended that the author properly absorb the reviewers' comments and make corresponding improvements and enhancements.”

We thank the Reviewer for the time and effort which they have dedicated to reviewing the manuscript, and for the constructive comments which they have provided. Our point-by-point responses now follow.

1.

“For the keywords, 'Carbon Intensity', 'Overnight Charging', 'Multi-Session', and 'Flexibility' should be added to attract a broader readership.”

Action: We have now added the recommended keywords to our manuscript.

2 (first part).

“Page 7, 'Electric Vehicles (EVs) are meeting the expectation of significantly reducing greenhouse gas emissions, as their Global Warming Potential (GWP), expressed in gCO2e/vkm (grams of carbon dioxide equivalent per vehicle-kilometre), is much lower than that of gasoline and diesel Internal Combustion Engine Vehicles (ICEVs) [1].' We cannot say electric vehicles are absolutely zero-carbon emissions. It depends on how the electricity used to charge the batteries is generated. If it relies on traditional thermal power, it still produces significant carbon emissions. A truly green electric vehicle should rely on excess electricity generated by renewable energy, storing it in the battery through the charging process. Therefore, the author needs to explain clearly the relationship and interaction mechanism between electric vehicles and renewable energy.”

We agree with the Reviewer that most of the emissions due to EVs depend on how the electricity used to charge the batteries is generated. Pleased see, below, a figure that illustrates this fact (taken from [1]) along with our elaboration on this matter.

The figure shows that production, maintenance and other minor aspects of the operation of an EV are approximately constant. In contrast, the electricity production cycle strongly depends on the specific country. This has a very low impact in countries with decarbonized energy, e.g. in France (FR, because of nuclear power) and Sweden (SE, because of hydro). Meanwhile, it has a high impact in countries which heavily leverage on higher-emissions fuels (e.g., coal). For instance, the total GWP of EVs in Estonia (EE) is higher than the average for ICEV-G or ICEV-D, due to the average mix from the grid. However, according to this report, the average GWP in Europe is already significantly lower than that of ICEV-G and ICEV-D, and it is supposed to decrease further in the future as the penetration level of energy generated from renewable resources is expected to increase.

Action: In the introduction of the revised manuscript, we have reported and elaborated this point (page 2, lines 10-17), thereby improving the explanation regarding the relationship and interaction mechanism between EV charging and renewable energy.

[1] Report, Study by Fraunhofer ISE & Fraunhofer ISI on behalf of Transport & Environment “Study on the “Potential of a full EV-power-system-integration in Europe & how to realise it””, 2023.

2 (second part).

“For example, most renewable energy sources are intermittent, opening spatial and temporal gaps between the availability of the energy and its consumption by end users. In order to address these issues, it is necessary to develop suitable energy storage systems for the power grid (10.1016/j.electacta.2019.03.056). Therefore, electric vehicles using lithium batteries have become an important support for the efficient recycling and utilization of renewable energy.”

We agree with the Reviewer on this point as well. In fact, our proposed methodology aims at charging the EVs mainly when carbon emissions are low, i.e. at times when more energy is generated from renewable sources. In this way, EVs would, indeed, be used for efficient recycling and utilization of renewable energy, as stated by the Reviewer.

Action: In the revised manuscript (Introduction, page 2, lines 20-24), we have added a comment to clarify that EVs may be used as mobile storage systems for the support of renewable energy utilization.

3.

“The presentation of innovative points requires enhanced comparison. The manuscript emphasizes "the first multi-period optimization for a single EV household scenario," but fails to adequately compare existing research. Therefore, it is recommended to include a literature comparison table in the Introduction, quantifying the marginal contribution of this study compared to single-period optimization (e.g., 24-hour strategies).”

In the revised manuscript, we have substantially extended our review of the state-of-the-art (Introduction, page 2, lines 37–59). We now clarify that while there is extensive literature on smart charging algorithms, most optimize functions of interest other than carbon emissions, such as cost of charging, load balancing, peak load, grid impact, or user convenience.

Studies which explicitly target carbon minimization remain comparatively limited, but we now cite and discuss several relevant contributions in the revised manuscript. For example, Dixon et al. [15] demonstrate that fleet-scale charge scheduling in Scotland can reduce emissions while mitigating wind curtailment; Huber et al. [16] and Mehlig et al. [17] highlight possible emission reductions based on average and marginal emission factors; Li et al. [18] propose a combined carbon emission and time-of-use price objective. These works highlight the growing research interest in controlled charging for carbon emission reduction, yet they typically assume perfect foresight, focus on fleet/aggregator scales, or they optimize only over a single charging session or day.

In contrast, our study addresses the under-explored case of domestic overnight charging with minimal hardware requirements (i.e. a simple smart plug), and it introduces a multi-session MPC framework that explicitly accounts for forecast uncertainty and user behaviour over several days. To make the comparison more transparent, we have also revised Table 2 (page 10) to include a column that quantifies the further contribution (in terms of emission reduction) of our multi-session strategy over single-period (24-hour) approaches.

Action: We have revised the Introduction section of our manuscript with an extended literature review (page 2, lines 37-59), now including additional references [16, 19]. Additionally, in Table 2 (page 10), we have added a column that numerically quantifies the marginal contribution of our multi-session optimization results compared to single-session optimization.

4.

“The theoretical basis for multi-period optimization is weak, and the reason for choosing a 7-day upper limit for the forecast window is not explained. Additional theoretical support is needed, including reference to time series forecasting theory and a sensitivity analysis demonstrating the saturation point between the window length and emission reduction benefits.”

As per the reviewer’s recommendation, we have provided additional results and discussion (Section 4.1), including a new figure (Fig. 4), to illustrate the saturation point between the forecast window length and emission reduction benefits. These results – which have been obtained via a Monte Carlo sensitivity analysis – demonstrate that increasing the future horizon length beyond 3-4 days does not provide significant improvement in terms of emission reductions. This can be attributed to the uncertainty in the forecasting of future carbon emissions. Thus, the arbitrary choice of a 7-day horizon is sufficient in the sense that it reveals the diminishing benefits of optimization horizons greater than 3-4 nights with respect to emission reduction.

Action: In the revised manuscript, we have added a new figure (see Fig. 4 of the revised manuscript) to evaluate how the emission reduction benefit varies with the choice of future horizon length. We have extended Section 4.1 to elaborate on the data presented in the new figure and to address the reviewer’s comment.

5.

“The full lifecycle analysis of the vehicle energy system needs improvement. The manuscript focuses solely on carbon emissions from the charging phase and fails to consider the impact of the lifespan degradation of the vehicle's energy storage device on the full lifecycle carbon footprint. It is recommended that the "Lifecycle Assessment" section include an analysis of the contribution of energy storage device (battery/fuel cell) durability to carbon emissions, while also introducing a state-of-health (SOH) degradation model to quantify the incremental carbon emissions implicit in long-term use. Authors can refer to 10.1109/TPEL.2024.3502499, which proposes a lifespan prediction framework under dynamic conditions that can be directly transferred to EV battery health management, helping authors construct a coupled SOH-carbon emissions model and enhancing the rigor of the full lifecycle analysis.”

In our work, we do not provide a full lifecycle carbon analysis of the energy storage device (such as the one in the figure above, in response to comment 2). We focus only on the carbon emissions from the charging phase (which are the most relevant for our analysis, being related to generation from renewable sources). Since we compare two different charging strategies strategies (i.e. plain charging and smart unidirectional charging), we assume that the difference in carbon emissions associated with the two strategies is only due to the charging choice.

The reviewer raises an important point, nonetheless: carbon emissions have direct and indirect sources. In our work, we only analyze the direct ones. However, as we will now explain in detail, our approach is also convenient in terms of further reducing indirect emissions.

Consider the following outcome of the report [1, page 70], where controlled unidirectional charging had been implemented to minimize the cost of charging: “the lifetime of the battery can be substantially extended by changing the target SOC upon departure from 100% to 80% instead. This increases the lifetime by an average of 40%. Controlled unidirectional charging increases the battery lifetime by a further 5-10% on average, compared to uncontrolled charging”.

In our work, we already set the target SOC upon departure to 80% for both scenarios (i.e. smart controlled and uncontrolled charging), specifically with the purpose of extending the lifetime of the battery.

However, as found in [1], smart control has the further advantage of increasing the battery lifetime by a further 5-10% on average (this result has been obtained by embedding both linear and non-linear ageing models, with better results with non-linear ageing models, as shown on page 90 of the report). Quoting from [1]: “the main source for extended lifetime is the reduced calendrical ageing resulting from more beneficial charging states”. Indeed, if uncontrolled, then charging is from a minimum level to a maximum level, while controlled charging operates more frequently around intermediate levels of the SOC.

Action: In section 2.1.2 (page 6, lines 158-160) of the revised manuscript, we cite reference [2] and discuss the findings there, to support our choice never to charge vehicles to more than 80% of the SOC. We also now highlight that further benefits may be obtained by applying our proposed strategy, specifically that smart charging strategies extend the lifetime of batteries, as explained in [2]. In addition, we clarify that more sophisticated models - such as the linear and nonlinear models of [2] - could be adopted, so as to take into account the degradation of the battery (page 6, lines 169-177).

[1] Report, Study by Fraunhofer ISE & Fraunhofer ISI on behalf of Transport & Environment “Study on the “Potential of a full EV-power-system-integration in Europe & how to realise it””, 2023.

6.

“The carbon intensity forecasting model is insufficiently disclosed. Formula (12) briefly mentions it as a "1-step-ahead forecast," but does not specify the specific forecasting algorithm (ARIMA/LSTM/Transformer), the input variables (whether covariates such as weather and electricity prices are considered), or the quantitative results of the forecast error (it is recommended to add MAPE/RMSE indicators).”briefly mentions it as a "1-step-ahead forecast," but does not specify the specific forecasting algorithm (ARIMA/LSTM/Transformer), the input variables (whether covariates such as weather and electricity prices are considered), or the quantitative results of the forecast error (it is recommended to add MAPE/RMSE indicators).”

In our study, we do not develop a carbon intensity forecasting model, but, instead, construct synthetic long-range forecasts from the publicly available UK National Energy System Operator (NESO) one-step-ahead forecasts. The NESO data provide actual carbon intensity values and corresponding one-step-ahead forecasts only. No multi-step forecasts are stored. To model the longer-range forecasts required by our optimization, we therefore synthesize multi-step-ahead forecasts by applying error growth rates estimated from the published CarbonCast benchmark study [22] (i.e. [20] in the original manuscript). This procedure allows us to account for the empirically observed growth of forecast uncertainty with horizon length, without assuming a specific forecasting model such as ARIMA, LSTM, or Transformer.

Action: We have revised Section 3.1 to emphasize that our objective in that section is not to provide a forecasting model, but, rather, to synthesize realistic long-range forecasts based on the available one-step ahead forecast sequence (NESO) and the published long-range error statistics (page 7, lines 197-199). Furthermore, we now include quantitative error statistics for both NESO-published one-step-ahead forecast errors (Fig. 2) and for the synthesized long-range forecast (Fig. 3).

7.

“The user behavior modeling is overly idealistic. The assumption that users "only request the required amount of electricity each day" fails to account for behavioral uncertainty, requiring the introduction of stochastic programming or robust optimization. The authors could further analyze the impact of temporary increases in user travel on emissions reductions, and appropriately cite behavioral economics research.”

We thank the Reviewer for the opportunity to clarify the misunderstanding here. We agree that the quoted phrase from the Abstract miscommunicated the behavioral modelling. In fact, our modelling does account for uncertainty in user behavior (specifically in the realized plug-in and plug-out times as well as in daily energy demand), as explained below.

We have extracted mean daily energy demand and day-to-day variability from real-world data [25] (which was [21] in the original manuscript). In our simulations, realized daily energy demand values are sampled from a gaussian distribution which is calibrated based on this real-world data, reflecting typical uncertainty in daily energy demand of EV drivers.

However, we then assume that the user communicates to the system a precautionary (upper bound) estimate of their typical daily energy requirement (corresponding to the 98th percentile value of actual energy demand). This ensures that expected energy requirements

---

## [Decision Letter · Decision Letter 1]

14 Jan 2026

Dear Dr. Wieberneit,

Thank you for submitting your manuscript to PLOS ONE. After careful consideration, we feel that it has merit but does not fully meet PLOS ONE’s publication criteria as it currently stands. Therefore, we invite you to submit a revised version of the manuscript that addresses the points raised during the review process.

We look forward to receiving your revised manuscript.

Kind regards,

Shih-Lin Lin, Ph.D

Academic Editor

PLOS One

Journal Requirements:

Reviewers' comments:

Reviewer's Responses to Questions

**Comments to the Author**

Reviewer #1: All comments have been addressed

Reviewer #2: All comments have been addressed

Reviewer #3: All comments have been addressed

Reviewer #4: All comments have been addressed

Reviewer #5: All comments have been addressed

Reviewer #6: All comments have been addressed

2. Is the manuscript technically sound, and do the data support the conclusions?

Reviewer #1: Partly

Reviewer #2: Yes

Reviewer #3: Yes

Reviewer #4: Yes

Reviewer #5: Yes

Reviewer #6: Yes

3. Has the statistical analysis been performed appropriately and rigorously?

Reviewer #1: N/A

Reviewer #2: Yes

Reviewer #3: Yes

Reviewer #4: No

Reviewer #5: Yes

Reviewer #6: Yes

4. Have the authors made all data underlying the findings in their manuscript fully available?

Reviewer #1: Yes

Reviewer #2: Yes

Reviewer #3: Yes

Reviewer #4: Yes

Reviewer #5: Yes

Reviewer #6: Yes

5. Is the manuscript presented in an intelligible fashion and written in standard English?

Reviewer #1: Yes

Reviewer #2: Yes

Reviewer #3: Yes

Reviewer #4: Yes

Reviewer #5: Yes

Reviewer #6: Yes

Reviewer #1: This paper proposes an innovative multi-stage MPC charging strategy to reduce the carbon intensity of overnight charging of electric vehicles. The experimental design is rigorous, and the results have practical value (e.g., a 37% carbon emission reduction). However, there is room for improvement in terms of hardware efficiency and future technology integration. Here are a few suggestions:

1. 1. The manuscript assumes constant charging efficiency, but energy losses (such as conversion losses) occur during the actual charging process, which are not modeled. In particular, the manuscript relies on "household smart plugs" as the hardware foundation, which may limit the practical applicability of the strategy. For example, compatibility issues may lead to user inconvenience or reduced efficiency.

It is recommended to briefly discuss the hardware constraints of wired charging (such as plug compatibility) in the "Introduction" or "Discussion" section. Recent research should be cited to emphasize how charging interoperability improves overall efficiency. For example, reference [10.1109/TPEL.2024.3429535] shows that optimizing wireless charging interoperability (such as defining impedance boundaries) can reduce energy losses, which is highly synergistic with the carbon reduction goals of this study. The quantitative methods proposed in the literature to assess charging efficiency can help demonstrate the indirect contribution of "hardware optimization" to carbon reduction.

2. Expand future research directions to include the potential of wireless charging. The current manuscript focuses on wired charging, but wireless charging is a key trend in the electrification transition. Ignoring this point may weaken the manuscript's long-term impact.

It is suggested to add a paragraph in the "Conclusion" or "Future Work" section to discuss the possibility of multi-technology integration. For example: Future research could explore the integration of intelligent scheduling strategies with wireless charging systems, where interoperability standards (Energies 16.4 (2023): 1653) can further improve user convenience and system efficiency. This literature summarizes the role of interoperability in promoting energy recovery, directly echoing the theme of this manuscript, "Utilizing Excess Electricity from Renewable Energy Sources."

Reviewer #2: My review comments have been fully addressed and resolved by the authors. They have carefully revised the manuscript according to the feedback, and the revised version satisfactorily answers all the concerns raised during the review process. The quality and clarity of the paper have been significantly improved as a result of these revisions. No further major issues remain to be discussed. Therefore, I believe the manuscript now meets the publication standards of the journal and can be accepted in its current form.

Reviewer #3: In this paper, authors presented a multi period predictive control based optimization approach for multiple nighttime electric vehicle charging stations with the goal of lowering carbon emissions. The experimental data is thorough, the paper's concept is straightforward, and the findings show notable decreases in carbon intensity. Considering all these points, the article is now recommended for acceptance.

Reviewer #4: The majority of electric vehicles (EVs) are charged domestically overnight, when the precise timing of power allocation is not important to the user, thus providing a source of flexibility that can be leveraged by charging control algorithms; This is addressed by the authors.

Dear Author,

I do appreciate your efforts for a well revision.

Best wishes,

Dr. V B Murali Krishna

Reviewer #5: The revised version of the paper is acceptable for publication . All the comments have been addressed by the authors .

Reviewer #6: This manuscript explores model predictive control for multi-session EV charging to minimize carbon emissions, leveraging UK grid data and user flexibility, which addresses a timely issue in sustainable mobility. The simulations demonstrate emission reductions, but the assumptions on user behavior and forecast accuracy are overly optimistic without sufficient sensitivity analysis. While valuable, the paper requires major revisions to enhance realism, expand comparisons, and refine the optimization framework prior to acceptance.

1. The introduction claims EVs reduce GWP significantly, but does not discuss variability across grids with high fossil fuel reliance. Elaborate on this with data from diverse regions to contextualize the UK focus.

2. The MPC formulation for multiple sessions is clear, but constraints on battery degradation are omitted. Incorporate state-of-health models to make the approach more practical.

3. The literature on smart charging is reviewed, but recent multi-period optimizations are underrepresented. To improve the methodological discussion and incorporate advanced flexibility strategies, consider these papers: Optimal scheduling of electric vehicle charging operations considering real-time traffic condition and travel distance, Expert Systems with Applications; Composite Neural Learning-Based Adaptive Actuator Failure Compensation Control for Full-State Constrained Autonomous Surface Vehicle, Neural Computing and Applications; Saturated-threshold event-triggered adaptive global prescribed performance control for nonlinear Markov jumping systems and application to a chemical reactor model, Expert Systems with Applications. This will help authors refine forecast windows and emission minimization.

4. Simulations use 2022 UK data, but lack scenarios with renewable variability like wind intermittency. Add stochastic elements to test robustness.

5. The 37% reduction is impressive, but not benchmarked against single-session MPC. Include direct comparisons to highlight multi-session advantages.

6. User flexibility assumptions (e.g., precautionary energy estimates) need empirical support. Reference surveys or add a section on incentive mechanisms.

7. The response to reviewers is addressed, but some figures still lack units or scales. Update all visuals for clarity.

.

Reviewer #1: No

Reviewer #2: No

Reviewer #3: **Yes:** Dr. Sanjiv Kumar KumarDr. Sanjiv Kumar KumarDr. Sanjiv Kumar KumarDr. Sanjiv Kumar Kumar

Reviewer #4: **Yes:** Dr. V B Murali KrishnaDr. V B Murali KrishnaDr. V B Murali KrishnaDr. V B Murali Krishna

Reviewer #5: **Yes:** Srinivasa Rao GampaSrinivasa Rao GampaSrinivasa Rao GampaSrinivasa Rao Gampa

Reviewer #6: No

---

## [Author Response · Author response to Decision Letter 2]

13 Feb 2026

Responses to Reviewer Comments

Reviewer #1:

This paper proposes an innovative multi-stage MPC charging strategy to reduce the carbon intensity of overnight charging of electric vehicles. The experimental design is rigorous, and the results have practical value (e.g., a 37% carbon emission reduction). However, there is room for improvement in terms of hardware efficiency and future technology integration. Here are a few suggestions:

1.1. The manuscript assumes constant charging efficiency, but energy losses (such as conversion losses) occur during the actual charging process, which are not modeled. In particular, the manuscript relies on "household smart plugs" as the hardware foundation, which may limit the practical applicability of the strategy. For example, compatibility issues may lead to user inconvenience or reduced efficiency.

It is recommended to briefly discuss the hardware constraints of wired charging (such as plug compatibility) in the "Introduction" or "Discussion" section. Recent research should be cited to emphasize how charging interoperability improves overall efficiency. For example, reference [10.1109/TPEL.2024.3429535] shows that optimizing wireless charging interoperability (such as defining impedance boundaries) can reduce energy losses, which is highly synergistic with the carbon reduction goals of this study. The quantitative methods proposed in the literature to assess charging efficiency can help demonstrate the indirect contribution of "hardware optimization" to carbon reduction.

Author Response

We thank the reviewer for highlighting charging-efficiency losses and practical implementation constraints. We agree that the original model did not take into account charging efficiency and did not explicitly represent conversion and auxiliary losses (EVSE, cable, and on-board charger). To address this, we have revised the formulation to include a charging-efficiency parameter \eta\in\left(0,1\right] in the SOC dynamics, capturing that only a fraction of the grid-supplied energy P\left[s,k\right]\Delta_t ends up stored in the battery.

Our contribution remains algorithmic: it is a multi-session MPC scheduling layer that minimizes emissions using carbon-intensity forecasts, and it is largely independent of the charging modality, provided charging power can be actuated (continuously via a controllable EVSE, or approximately via time-averaged on/off switching). Under the (standard) approximation that \eta is approximately constant for domestic AC charging over the considered SOC range (20–80%) and common across the compared strategies, including \eta primarily rescales SOC evolution and absolute energy flows but does not materially change the relative comparison between uncontrolled charging, single-session MPC, and multi-session MPC.

Regarding the suggested reference: We agree that charging hardware interoperability affects losses during the charging process, however, given that the present work deliberately focusses on the temporal control dimension, we feel that a detailed discussion of impedance-space interoperability for wireless systems would be out of scope. However, we appreciate the broader point that interoperability and hardware efficiency are complementary levers to reduce lifecycle emissions.

Author Action

We have updated the SOC charging dynamics (Eq. 3) to include a charging-efficiency parameter \eta, and added \eta to the notation table.

We have included a short remark (Model Accuracy / Practical Implementation) stating that we neglect detailed power-electronics/thermal effects and treat \eta as approximately constant; noting that more detailed efficiency models can be incorporated without changing the proposed control framework.

We have added a brief clarification that the control approach is hardware-agnostic, requiring only an interface to actuate charging power (or its time-averaged equivalent via on/off control), and that the reported comparisons assume the same \eta across strategies.

2. Expand future research directions to include the potential of wireless charging. The current manuscript focuses on wired charging, but wireless charging is a key trend in the electrification transition. Ignoring this point may weaken the manuscript's long-term impact.

It is suggested to add a paragraph in the "Conclusion" or "Future Work" section to discuss the possibility of multi-technology integration. For example: Future research could explore the integration of intelligent scheduling strategies with wireless charging systems, where interoperability standards (Energies 16.4 (2023): 1653) can further improve user convenience and system efficiency. This literature summarizes the role of interoperability in promoting energy recovery, directly echoing the theme of this manuscript, "Utilizing Excess Electricity from Renewable Energy Sources."

Author Response

We agree that wireless charging technologies are a relevant long-term development in EV infrastructure and that interoperability standards for such systems can enhance user convenience and system efficiency. Our MPC framework is agnostic to the physical interface of the charging system; it optimizes power trajectories given availability windows and forecasted carbon intensity. In principle, the same logic could be applied to wireless systems.

At the same time, the present paper is deliberately scoped to a very simple and widely deployable use case—domestic overnight charging via a conventional smart plug. We therefore prefer to keep wireless charging clearly identified as future work, rather than expanding the technical content in that direction.

Author Action

In the Conclusions we added a short paragraph stating that:

Although the case study is scoped for domestic charging with smart plugs, the proposed multi-session MPC framework is hardware agnostic and could be interfaced with other technologies such as wireless charging.

Reviewer #2:

My review comments have been fully addressed and resolved by the authors. They have carefully revised the manuscript according to the feedback, and the revised version satisfactorily answers all the concerns raised during the review process. The quality and clarity of the paper have been significantly improved as a result of these revisions. No further major issues remain to be discussed. Therefore, I believe the manuscript now meets the publication standards of the journal and can be accepted in its current form.

Author Response

We thank the reviewer for their feedback and are pleased that the revision addressed the earlier concerns.

Reviewer #3:

In this paper, authors presented a multi period predictive control based optimization approach for multiple nighttime electric vehicle charging stations with the goal of lowering carbon emissions. The experimental data is thorough, the paper's concept is straightforward, and the findings show notable decreases in carbon intensity. Considering all these points, the article is now recommended for acceptance.

Author Response

We thank the reviewer for their feedback and are pleased that the revision addressed the earlier concerns.

Reviewer #4:

The majority of electric vehicles (EVs) are charged domestically overnight, when the precise timing of power allocation is not important to the user, thus providing a source of flexibility that can be leveraged by charging control algorithms; This is addressed by the authors.

Dear Author,

I do appreciate your efforts for a well revision.

Best wishes,

Dr. V B Murali Krishna

Author Response

We thank the reviewer for their feedback and are pleased that the revision addressed the earlier concerns.

Reviewer #5:

The revised version of the paper is acceptable for publication . All the comments have been addressed by the authors .

Author Response

We thank the reviewer for their feedback and are pleased that the revision addressed the earlier concerns.

Reviewer #6:

This manuscript explores model predictive control for multi-session EV charging to minimize carbon emissions, leveraging UK grid data and user flexibility, which addresses a timely issue in sustainable mobility. The simulations demonstrate emission reductions, but the assumptions on user behavior and forecast accuracy are overly optimistic without sufficient sensitivity analysis. While valuable, the paper requires major revisions to enhance realism, expand comparisons, and refine the optimization framework prior to acceptance.

1. The introduction claims EVs reduce GWP significantly, but does not discuss variability across grids with high fossil fuel reliance. Elaborate on this with data from diverse regions to contextualize the UK focus.

Author Response

We agree that grid carbon intensity varies widely across regions and that this affects both the absolute and relative lifecycle benefits of EVs. In the revised introduction we already expanded the discussion to emphasise that:

Use-phase GWP is highly contingent on the generation mix, and the same EV can have very different lifecycle GWP depending on local grid carbon intensity;

Some countries/regions with predominantly fossil-based generation (e.g., Estonia) show much higher use-phase GWP than predominantly fossil fuel grids (e.g., France or Sweden).

To respond more explicitly to the reviewer’s concern, we now:

Clarify that the UK is used as a case study with rich, publicly available half-hourly carbon intensity data;

Note that achievable emission reductions are contingent on the specific carbon intensity sequence of the electricity grid.

Author action

In the Introduction, we refined the paragraphs discussing lifecycle GWP and grid carbon intensity to:

Explicitly highlight the variability between low-carbon and high-fossil grids;

Clarify that the UK is used because of the availability of high-resolution carbon intensity data;

State that the same MPC framework can be applied to grids with different carbon profiles, so long as the carbon intensity is not constant, with absolute savings depending on the average carbon intensity and its variability over time.

2. The MPC formulation for multiple sessions is clear, but constraints on battery degradation are omitted. Incorporate state-of-health models to make the approach more practical.

Author response

We agree that explicit state-of-health modelling is important for long-term operational planning and that there is a natural coupling between carbon-optimal charging and degradation-optimal charging. In this work, we adopt a simple surrogate for battery state of health:

SOC is constrained to [20%, 80%], in line with recent reports indicating that limiting SOC to this range can substantially extend battery lifetime;

We focus on the relative difference between uncontrolled and smart charging under this SOC policy.

We have already acknowledged in a Remark (Model Accuracy) and in the Conclusions that we neglect SOC-dependent efficiency and battery ageing, and cited recent work showing that smart charging can extend battery lifetime. In response to the reviewer’s request, we now emphasise more clearly that:

Our SOC constraints are chosen to be consistent with reduced degradation;

Integrating an explicit SoH model into the MPC (e.g., via additional state dynamics or cost terms) is an important but non-trivial extension and is left to future work to avoid over-complicating the current framework.

Author action

In Section 2.1.2, in the Remark (Model Accuracy), we added explicit clarification that:

SOC bounds [20%, 80%] are chosen to reflect commonly recommended operating ranges for reduced degradation.

Ageing and detailed SoH dynamics are not modelled in the current optimisation.

In the Conclusions, we strengthened the paragraph discussing battery lifetime to:

Highlight that maintaining SOC in reduced ranges may lower indirect emissions by delaying battery replacement.

Explicitly identify the integration of coupled SoH–carbon models within the MPC as a key avenue for future work.

3. The literature on smart charging is reviewed, but recent multi-period optimizations are underrepresented. To improve the methodological discussion and incorporate advanced flexibility strategies, consider these papers: Optimal scheduling of electric vehicle charging operations considering real-time traffic condition and travel distance, Expert Systems with Applications; Composite Neural Learning-Based Adaptive Actuator Failure Compensation Control for Full-State Constrained Autonomous Surface Vehicle, Neural Computing and Applications; Saturated-threshold event-triggered adaptive global prescribed performance control for nonlinear Markov jumping systems and application to a chemical reactor model, Expert Systems with Applications. This will help authors refine forecast windows and emission minimization.

Author response

We thank the reviewer for suggesting additional references.

After reviewing the proposed papers, we decided not to include them as citations because they do not directly inform the modelling assumptions or optimisation structure used in this manuscript.

The EV scheduling paper (Expert Systems with Applications: “Optimal scheduling of electric vehicle charging operations considering real-time traffic condition and travel distance”) studies urban/public-charging operations with routing/traffic and queueing/assignment decisions, with the objective of minimizing total charging time. Charging power is treated as a fixed parameter rather than a controllable trajectory, and the formulation does not address carbon-intensity–driven objectives or multi-session domestic overnight charging. We therefore believe citing it would not materially clarify our MPC formulation, forecast-window design, or emissions-minimisation focus.

The other two suggested papers address adaptive/event-triggered control in unrelated domains (autonomous surface vehicles and nonlinear Markov jumping systems applied to a chemical reactor). While methodologically interesting, their control architectures and application contexts are not used here and would be considered irrelevant to the smart-charging literature reviewed in this work.

Author action

Instead of adding these citations, we strengthened the Introduction/Literature discussion to more explicitly position our contribution relative to multi-period EV charging optimisation: we clarify that our novelty is a multi-session, carbon-aware MPC scheduling layer for domestic overnight charging, and we avoid expanding the scope to routing/queueing-based public-charging formulations.

4. Simulations use 2022 UK data, but lack scenarios with renewable variability like wind intermittency. Add stochastic elements to test robustness.

Author Response

We appreciate this important point on robustness to renewable variability. In our study, the carbon-intensity input is the half-hourly historical UK grid carbon intensity for 2022, which already embeds real renewable variability (including wind intermittency) and its interaction with demand and conventional generation. In addition, for the annual results we perform multiple Monte Carlo runs with randomised user sequences (plug-in times and daily energy consumption) and report mean and standard deviation across these runs. The controller is therefore evaluated under both realistic grid variability and behavioural variability.

We agree that this was not sufficiently emphasised in the original manuscript, and we are grateful for the suggestion to make the robustness aspect more explicit.

Author action

We have revised Section 3 to clarify that:

(i) the simulations use full-year, half-hourly historical UK carbon-intensity data, which inherently captures renewable intermittency, and especially wind intermittency; and

(ii) all annual results are based on multiple Monte Carlo runs with randomised user behaviour, with mean and standard deviation reported as a robustness indicator.

5. The 37% reduction is impressive, but not benchmarked against single-session MPC. Include direct comparisons to highlight multi-session advantages.

Author response

We agree that explicitly highlighting the benefit of multi-session MPC over single-session MPC is important. In the current results section and Table 2, we alr

---

## [Decision Letter · Decision Letter 2]

10 Mar 2026

Multi-Session Smart Night Charging of EVs for Accelerated Decarbonization of Electric Mobility

PONE-D-25-37660R2

Dear Dr. Wieberneit,

We’re pleased to inform you that your manuscript has been judged scientifically suitable for publication and will be formally accepted for publication once it meets all outstanding technical requirements.

No further revisions are required before publication. In addition, the additional citation suggested by Reviewer 6 is not required for acceptance.

Congratulations, and we look forward to seeing your work published.

Kind regards,

Shih-Lin Lin, Ph.D

Academic Editor

PLOS One

Additional Editor Comments (optional):

Reviewers' comments:

Reviewer's Responses to Questions

**Comments to the Author**

Reviewer #1: All comments have been addressed

Reviewer #6: (No Response)

2. Is the manuscript technically sound, and do the data support the conclusions?

Reviewer #1: Partly

Reviewer #6: (No Response)

3. Has the statistical analysis been performed appropriately and rigorously?

Reviewer #1: N/A

Reviewer #6: (No Response)

4. Have the authors made all data underlying the findings in their manuscript fully available?

Reviewer #1: Yes

Reviewer #6: (No Response)

5. Is the manuscript presented in an intelligible fashion and written in standard English?

Reviewer #1: Yes

Reviewer #6: (No Response)

Reviewer #1: I have carefully reviewed the updated version of the manuscript, and I consider the authors have made great efforts to improve the quality of the manuscript. I consider the current version of the manuscript can be accepted for the final publication. Hence, no further comments are given at this stage.

Reviewer #6: The authors have not yet justified the contributions of their paper. What is the actual improvement with respect to other existing related results? It is necessary to additionally improve the basis of the contribution in relation to more recent references. Some recent relevant contributions appear to have been missed. The literature review on the topic is not thorough, some related results, proposed in the previous review round, should be included in this paper, as well as: Relaxed Model Predictive Control of T-S Fuzzy Systems via A New Switching-Type Homogeneous Polynomial Technique, IEEE Transactions on Fuzzy Systems, give a short comment and in that way, point out other approaches and possibilities.

.

Reviewer #1: No

Reviewer #6: No

---

## [Editor Report · Acceptance letter]

PONE-D-25-37660R2

PLOS One

Dear Dr. Wieberneit,

I'm pleased to inform you that your manuscript has been deemed suitable for publication in PLOS One. Congratulations! Your manuscript is now being handed over to our production team.

Kind regards,

on behalf of

Professor Shih-Lin Lin

Academic Editor

PLOS One